# Single-cell analysis reveals prognostic fibroblast subpopulations linked to molecular and immunological subtypes of lung cancer

Christopher J. Hanley [1,2] ✉, Sara Waise [1,7], Matthew J. Ellis[1,7], Maria A. Lopez[3], Wai Y. Pun[3], Julian Taylor[3], Rachel Parker[1], Lucy M. Kimbley [1], Serena J. Chee[1,4], Emily C. Shaw[3], Jonathan West [1,5], Aiman Alzetani [6], Edwin Woo[6], Christian H. Ottensmeier [1,2,4], Matthew J. J. Rose-Zerilli[1,5] & Gareth J. Thomas [1,2,3] ✉

Fibroblasts are poorly characterised cells that variably impact tumour progression. Here, we use single cell RNA-sequencing, multiplexed immunohistochemistry and digital cytometry (CIBERSORTx) to identify and characterise three major fibroblast subpopulations in human non-small cell lung cancer: adventitial, alveolar and myofibroblasts. Alveolar and adventitial fibroblasts (enriched in control tissue samples) localise to discrete spatial niches in histologically normal lung tissue and indicate improved overall survival rates when present in lung adenocarcinomas (LUAD). Trajectory inference identifies three phases of control tissue fibroblast activation, leading to myofibroblast enrichment in tumour samples: initial upregulation of inflammatory cytokines, followed by stress-response signalling and ultimately increased expression of fibrillar collagens. Myofibroblasts correlate with poor overall survival rates in LUAD, associated with loss of epithelial differentiation, *TP53* mutations, proximal molecular subtypes and myeloid cell recruitment. In squamous carcinomas myofibroblasts were not prognostic despite being transcriptomically equivalent. These findings have important implications for developing fibroblast-targeting strategies for cancer therapy.

Cancer-associated fibroblasts (CAFs) are a prominent component of solid tumours, associated with poor prognosis in a wide range of cancer types[1]. Multiple studies have shown their positive influence on tumour progression: promoting metastasis, immune evasion and therapy resistance, making them an attractive therapeutic target. However, multiple CAF targeting strategies have failed in the clinic[2,3], and it is increasingly clear that CAFs play an important but context-dependent role in cancer, with subpopulations capable of both promoting and suppressing tumour progression[4–6]. This suggests that a more nuanced understanding of fibroblast heterogeneity and biology is required if CAF targeting is to become an effective part of cancer therapy.

Two major CAF subpopulations have been described in several cancer types: myofibroblastic-CAFs (myoCAFs) and inflammatory-CAFs (iCAFs)[7,8]. The role of these CAF subpopulations remains poorly

[1]School of Cancer Sciences, University of Southampton, Southampton SO16 6YD, UK. [2]Cancer Research UK and NIHR Southampton Experimental Cancer Medicine Centre, Southampton SO16 6YD, UK. [3]Department of Histopathology, University Hospital Southampton NHS Foundation Trust, Southampton SO16 6YD, UK. [4]Institute of Systems, Molecular and Integrative Biology (ISMIB) and Liverpool Experimental Cancer Medicines Centre, University of Liverpool, Liverpool L69 7BE, UK. [5]Institute for Life Sciences, University of Southampton, Southampton SO17 1BJ, UK. [6]Department of Thoracic surgery, University Hospital Southampton NHS Foundation Trust, Southampton SO16 6YD, UK. [7]These authors contributed equally: Sara Waise, Matthew J. Ellis. ✉e-mail: C.J.Hanley@soton.ac.uk; G.Thomas@soton.ac.uk

understood and may depend on further sub-categorisations, as described in breast cancer[7]. A key area of contention is myoCAF's role in tumour progression, with conflicting reports produced in different tumours. Multiple human tissue analyses have shown that α-smooth muscle actin (SMA)-positive myoCAFs are linked to poor prognosis in many cancer types[1] and to immunotherapy resistance in non-small cell lung cancer (NSCLC)[7]. However, in pancreatic ductal adenocarcinoma (PDAC) models, these cells can suppress tumour progression[4–6,9]. Murine studies have also shown that stromal cells expressing iCAF markers increase chemoresistance, metastases and immune suppression[10,11]. However, iCAF's influence on human tumour progression is yet to be robustly investigated. Importantly, myoCAF and iCAF are now recognised to be plastic cell populations interconvertible in vitro depending on the biochemical and mechanical features of the culture environment[9].

A significant barrier to understanding CAF biology is the paucity of well-characterised markers and lack of uniform terminology to describe different phenotypes. Technological advances, such as single-cell RNA-sequencing (scRNA-seq), are expanding the understanding of fibroblast heterogeneity. These techniques have been used to identify multiple fibroblast subpopulations in different tissue and cancer types[8,12–15]. However, this remains an evolving area of research with multiple discrepancies. For example, the degree of fibroblast heterogeneity (number of subpopulations identified) is commonly confounded across studies due to varying resolution of sub-clustering and whether mural cells (vascular smooth muscle cells and pericytes) are excluded prior to analysis[12,16]. Additionally, sub-population marker genes are typically calculated based on single-cell expression profiles without confirming statistical significance at a sample-level, leading to limited consistency across studies[12,16]. Furthermore, there has been minimal examination of how CAF phenotypes differ from healthy tissue counterparts; how comparable fibroblast phenotypes are across disease states; and how specific subpopulations impact disease progression.

Considerable research has been performed using scRNA-seq to characterise fibroblast heterogeneity in control lung tissues and interstitial lung diseases[13–15,17,18]. These studies have progressed from initial characterisations solely at a transcriptomic level[13,17], to the establishment of orthogonally validated subpopulations associated with specific niches in lung tissue[14,15]. This has culminated in the consistent identification of alveolar, adventitial and myofibroblast subpopulations in human and murine lung tissue[14,18]. A recent study by Buechler et al. provided further context to fibroblast heterogeneity across multiple human and murine tissues: identifying the adventitial subpopulation as a 'universal' fibroblast phenotype present across all tissues analysed; and a variety of tissue or pathology-specific subpopulations[19].

The robust characterisation of fibroblast phenotypes in non-cancerous lung tissues provides an excellent baseline for investigating how lung cancer impacts fibroblast phenotypes. Non-small cell lung cancer (NSCLC), the most common form of lung cancer, consists of multiple histological subtypes (adenocarcinomas [LUAD], squamous cell carcinomas [LUSC] and large cell carcinomas[20,21]), which are further subcategorised into molecular[22,23] and morphological[24,25] subtypes that link to patient survival. NSCLCs also have a high level of stromal and immune cell infiltration as well as high tumour mutational burden[26]. Therefore, we hypothesise that NSCLCs would exhibit a high degree of fibroblast heterogeneity and represent a suitable model to broadly characterise CAF phenotypes.

Here we use transcriptomics, multiplexed immunohistochemistry (MxIHC) and digital cytometry (implemented using CIBERSORTx[27]) to examine fibroblast heterogeneity in human NSCLC and control lung tissue: investigating phenotypic diversity between and within the major fibroblast subpopulations; comparing these phenotypes across multiple tissue types; examining their spatial distribution; and assessing their clinical significance across multiple NSCLC cohorts.

## Results

### In silico fibroblast identification from scRNA-seq data

We performed scRNA-seq on human lung tissue samples ($n = 18$; six control, seven squamous cell carcinomas [LUSC], and five adenocarcinomas [LUAD]; Fig. 1a and Supplementary Data 1), using a previously-described protocol to enrich for fibroblasts during tissue disaggregation[28]. Given that samples were obtained from surgical resections and not processed simultaneously, we assessed whether this generated batch effects in the scRNA-seq data using k-nearest neighbour overlap across cells collected from individual patients. This identified significantly increased overlap, which would therefore impact clustering results (Supplementary Fig. 1a). To mitigate this, we applied a reciprocal PCA (rPCA) based data integration (Supplementary Fig. 1a). Clustering and differential gene expression analyses were then performed. This identified multiple distinct mesenchymal, immune and epithelial cell populations (Fig.1c and Supplementary Fig. 1b–d). The mesenchymal cells included two separate clusters: endothelial cells (marked by *VWF* among other canonical markers) and stromal cells (marked by *DCN* and *DPT*) (Fig.1c and Supplementary Fig. 1e).

To investigate fibroblasts, we examined the stromal cell cluster. Given that mural cells are a prominent stromal cell type in lung tissue[18] and were not identified in our original clustering, we sought to determine whether the stromal cell cluster contained both fibroblasts and mural cells. This is important since myofibroblasts are commonly identified by the expression of genes encoding proteins involved in cellular contraction (e.g. *ACTA2* [gene encoding αSMA])[29], which are highly expressed in mural cells. To differentiate between these cells, we identified genes differentially expressed between fibroblasts and mural cells in human lung tissues using scRNA-seq data from a human lung cell atlas[18] (HLCA; Fig. 1e). We restricted these markers to human homologues of genes previously described to demarcate fibroblasts and mural cells in multiple murine organs[30], generating consensus gene signatures for fibroblasts and mural cells (Fig. 1e). To determine whether these signatures could effectively differentiate mural cells from fibroblasts, we calculated their average per-cell expression in the HLCA dataset, showing 99% accuracy for identifying the two cell-types (Fig. 1f, g). We then examined the expression of these signatures in our dataset (Fig. 1h–j), demonstrating that both mural cells ($n = 69$) and fibroblasts ($n = 885$) were detected within the stromal cell cluster. To further test whether fibroblasts and mural cells commonly cluster together when analysing whole-tissue homogenate tumour samples by scRNA-seq, we repeated this analysis on multiple publicly available datasets[12,16,31,32] consistently observing similar results (representative example shown in Supplementary Fig. 1f, g).

In summary, we have identified a broadly applicable method for distinguishing fibroblasts from mural cells in scRNA-seq datasets, demonstrating that this is a critical step in the characterisation of fibroblast (or mural cell) heterogeneity within the tumour microenvironment.

### Three fibroblast subpopulations are present in NSCLC

It is challenging to generate a scRNA-seq dataset that contains sufficient samples to enable population level (across multiple patients) characterisation of fibroblast subpopulations, due to costs, sample availability and difficulties in isolating these cells from tissues leading to variable proportions across tissues[12,28]. To overcome this challenge and comprehensively examine fibroblast heterogeneity in NSCLC, we repeated the process of in silico fibroblast sorting (excluding mural cells) with six further scRNA-seq datasets from human NSCLC and control tissue samples[16,18,31–34]. This generated a dataset with 9673 fibroblasts (Fig. 2a; including 5183 from 39 control tissues; 3440 from 46 LUAD samples and 654 from 16 LUSC samples). To integrate and correct batch effects between datasets, we used canonical correlation analysis (Supplementary Fig. 2a)[35]. We then performed unsupervised clustering, using a shared nearest neighbour modularity optimisation

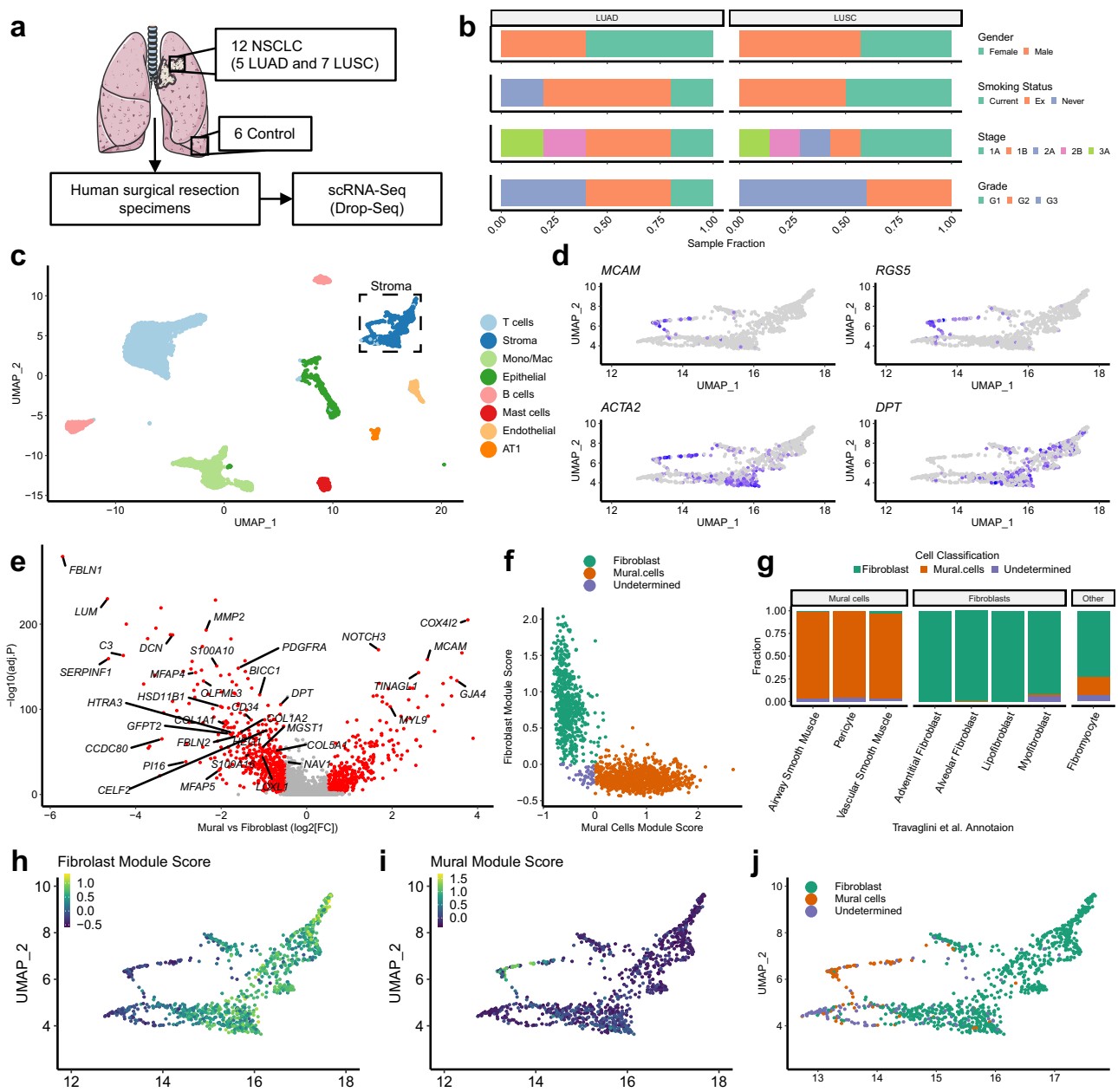

**Fig. 1 | Fibroblast identification through single-cell RNA-sequencing analysis of whole-tissue homogenates derived from human NSCLC tumour samples.** **a** Schematic illustrating sample processing and analysis methodology used to generate the target lung drop-seq (TLDS) dataset, comprised of human control ($n = 6$) and NSCLC ($n = 12$) samples. The Figure was partly generated using Servier Medical Art, provided by Servier, licensed under a Creative Commons Attribution 3.0 unported license. **b** Barplots showing key demographics of the TLDS dataset, further details provided in Supplementary Data 1. **c.** 2D visualisation (UMAP dimensionality reduction) of pooled data from all samples' whole-tissue scRNA-seq data, highlighting different cell types. Further analysis is shown in Supplementary Fig. 1a–d. AT1 alveolar type 1 cells, Mono/Mac monocytes and macrophages. **d** Feature plots showing the expression of canonical markers for mural cells (*MCAM* and *RGS5*), mural cells and myofibroblasts (*ACTA2*) and fibroblasts (*DPT*), in the

stromal cell cluster (subset from panel **c**). **e** Volcano plot showing genes differentially expressed (Bonferroni adj.$P < 0.01$ and absolute logFC $>0.5$, shown in red) between mural cells and fibroblasts from a recently published human lung cell atlas[18] (HLCA). Genes included in the consensus gene signatures identified for mural cells or fibroblasts are labelled. **f** Scatter plot showing the classification of fibroblasts and mural cells based on consensus gene signatures (defined in **e**), applied to stromal cells from the HLCA dataset[18]. **g** Barplots showing the results of the gene signature classification results compared to the original HLCA publication's cell type annotation. **h** Feature plot showing the expression of consensus fibroblast marker gene signature in the TLDS dataset. **i** Feature plot showing the expression of consensus mural marker gene signature in the TLDS dataset. **j** UMAP plot showing the result of fibroblast and mural cell demarcation in the TLDS dataset using the consensus gene signature approach.

algorithm[36]. To determine the most biologically informative clustering solution, we ran this analysis varying the resolution parameter to iteratively increase the number of clusters identified and examined the number of marker genes identified for each cluster (sample-level average log fold change >1, adj.$P < 0.01$ and expressed by a minimum of 50% of samples). This showed that three major clusters were

consistently identified, whereas higher resolution clustering led to the identification of clusters with no or very few marker genes meeting these criteria (Fig. 2b and Supplementary Fig. 2b).

The three clusters identified were consistent with fibroblast subpopulations previously described in control lung tissue[14,18] (adventitial, alveolar and myofibroblasts; Supplementary Fig. 2c–e). To

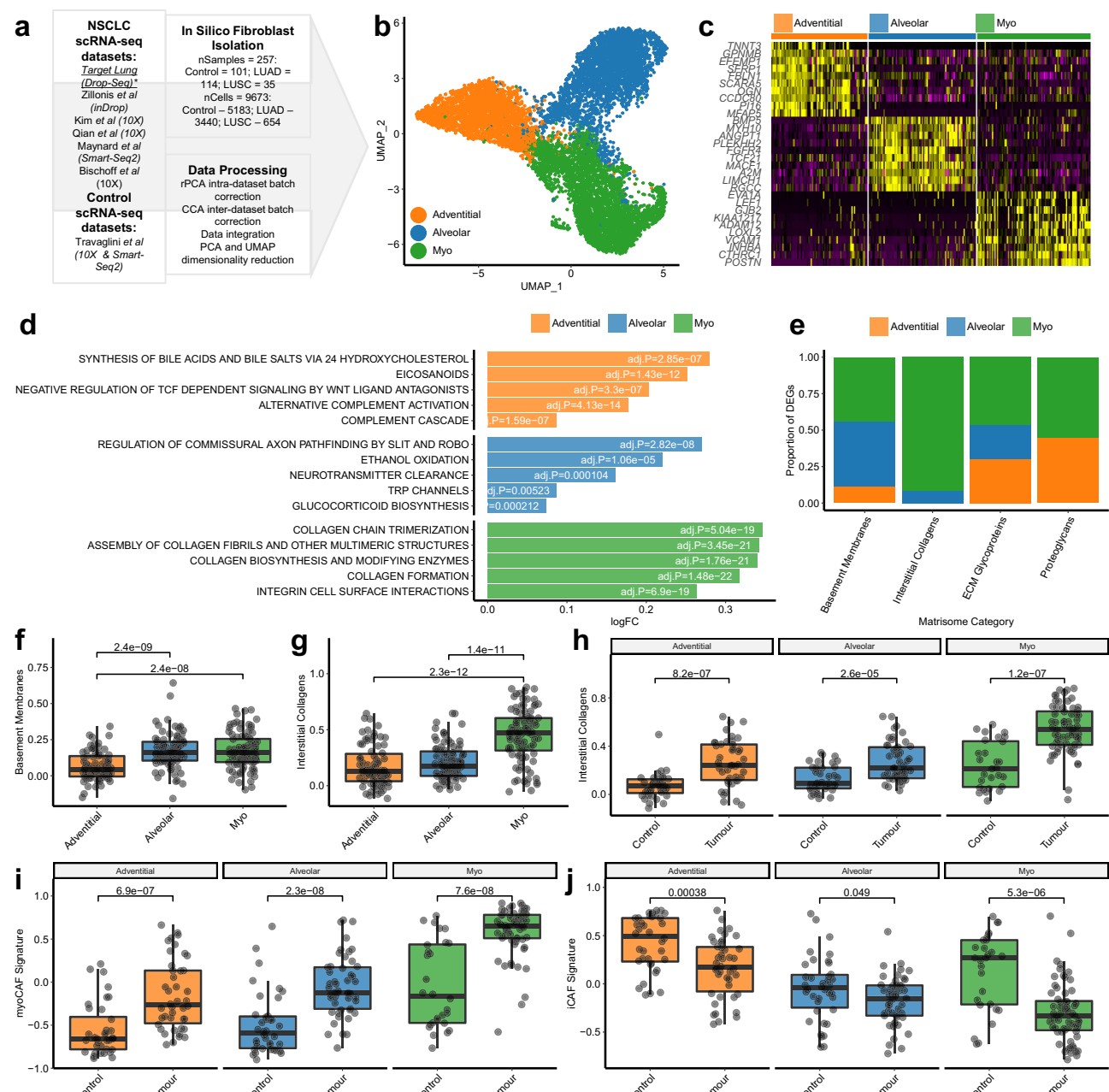

**Fig. 2 | Investigating fibroblast heterogeneity in NSCLC through the integration of seven scRNA-seq datasets. a** Schematic overview of the data processing pipeline implemented to generate an integrated dataset for analysing fibroblast's transcriptomic heterogeneity in NSCLC. **b** 2D visualisation (UMAP dimensionality reduction) of fibroblast transcriptomes, highlighting the three major subpopulations identified through unsupervised clustering. Further analysis is shown in Supplementary Fig. 1a–e. **c** Heatmap showing the sample-level expression (averaged over single cells) for the ten most significant markers of each subpopulation. Complete differential expression results are provided in Supplementary Data 2. **d** Bar plot showing the log2 fold change for the most significantly upregulated REACTOME pathways in each subpopulation, calculated through GSVA and Empirical Bayes Statistics for differential expression (exact Bonferroni adjusted *p* values are also shown). Complete results from this analysis are provided in Supplementary Data 3. **e** Bar plot showing the proportion of different matrisome components differentially expressed by each subpopulation. **f** Boxplot showing sample-level expression (averaged over single cells) of genes encoding basement membrane components for each subpopulation. Nominal *p* values for the Wilcoxon signed-ranks test are also shown(*n* = 78 [adventitial], 87 [alveolar] and 92 [myo]).

**g** Boxplot showing sample-level expression (averaged over single cells) of genes encoding interstitial collagens for each subpopulation. Nominal *p* values for the Wilcoxon signed-ranks test are also shown (*n* = 78 [adventitial], 87 [alveolar] and 92 [myo] independent samples). **h** Boxplot showing sample-level expression (averaged over single cells) of genes encoding interstitial collagens for each subpopulation split by tissue type. Nominal *p* values for the Wilcoxon signed-ranks test are also shown (*n* = 36/42 [adventitial, control/tumour], 38/49 [alveolar], 28/64 [myo]). **i** Boxplot showing sample-level expression (averaged over single cells) of genes encoding myoCAF markers for each subpopulation split by tissue type. Nominal *p* values for the Wilcoxon signed-ranks test are also shown (*n* as per panel **h**). **j** Boxplot showing sample-level expression (averaged over single cells) of genes encoding iCAF markers for each subpopulation split by tissue type. Nominal *p* values for the Wilcoxon signed-ranks test are also (*n* as per panel **h**). All statistical tests carried out were two-sided and boxplots are displayed using the Tukey method (centre line, median; box limits, upper and lower quartiles; whiskers, last point within a 1.5x interquartile range). Source data for panels **f**–**j** are provided in the Source Data file.

comprehensively characterise these fibroblast subpopulations at a population level, we first calculated sample-level gene expression profiles for each subpopulation (averaged over single cells) and then performed differential gene expression analysis (Fig. 2c and Supplementary Data 2); gene set variation analysis (GSVA) using the REACTOME pathways (Fig. 2d and Supplementary Data 3) and GSVA using gene signatures for previously described fibroblast subpopulations (Supplementary Fig. 2f and Supplementary Data 4).

Differential expression analysis using single-cell data identified 622 genes upregulated by myofibroblasts, 188 of which remained significant (adj.$P < 0.01$) in sample-level analysis (Supplementary Data 2). This attrition in marker identification from single-cell analysis to sample-level analysis demonstrates the importance of ensuring that scRNA-seq datasets are adequately powered in terms of sample numbers to detect population-level variance. Key markers for the myofibroblast subpopulation included *MMP11, POSTN, CTHRC1, COL1A1, ACTA2* and *COL3A1*. GSVA showed that these cells significantly upregulate multiple pathways involved in generating collagenous ECM (Fig. 2d), consistent with the well-described role played by myofibroblasts in fibrosis and myoCAF in cancer[37,38]. In addition to these pathways involving ECM biosynthesis, multiple pathways involving cell-ECM interactions were also upregulated, including integrin cell surface interactions and syndecan interactions (Fig. 2d and Supplementary Data 3). These cells also upregulated multiple previously described myofibroblast and myoCAF gene signatures (Supplementary Fig. 2f and Supplementary Data 4).

Differential expression analysis using single-cell data identified 481 genes upregulated in adventitial fibroblasts compared to the other subpopulations, 73 of these genes remained significant when analysing differential expression across samples—including *PI16, IGFBP6, MFAP5, APOD, PLA2G2A* and *GSN* (Supplementary Data 2). GSVA identified increased expression of multiple pathways involving *PTGIS* (Prostaglandin I2 synthase), including synthesis of prostaglandins and bile acids/salts (Fig. 2d), critical to the mobilisation of cholesterol from lung phagocytes[39]. The alternative complement activation pathway was also significantly upregulated, involving *C3* and *CFD* (Fig. 2d). Adventitial fibroblasts also upregulated gene signatures associated with the *COL14A1* + matrix fibroblasts described by ref. [16]; the *PI16* + 'universal' fibroblast population described by ref. [19]; and iCAF subpopulations described in pancreatic and breast cancer[7,8] (Supplementary Fig. 2f and Supplementary Data 4).

For alveolar fibroblasts, single-cell differential expression analysis identified 672 upregulated genes, 78 of which remained significant in sample-level analysis -including *MACF1, RGCC, INMT, LIMCH1, A2M* and *GPC3* (Supplementary Data 2). GSVA identified upregulation of TRP channels (Fig. 2d), which detect and transduce sensory signals (e.g. oxidative stress, pH and heat) into chemical or electrical signals to regulate cellular responses[40]. Pathways involving SLIT and ROBO gene family members were also upregulated (Fig. 2d), which have well-described roles in regulating commissural axon pathfinding but have also been shown to regulate fibroblast migration in arthritis[41]. Alveolar fibroblasts also upregulated gene signatures associated with the *COL13A1* + matrix fibroblast population described by ref. [16] and the lung-specific *NPNT* + fibroblast population described by ref. [19] (Supplementary Fig. 2f and Supplementary Data 5).

Production and remodelling of the ECM is a key function of fibroblasts across all tissues[42]. Consistent with this, the genes differentially expressed between fibroblast subpopulations were significantly enriched in components of the ECM (Fisher's exact $p = 1.34e-62$), with 31% of all differentially expressed genes associated with the matrisome[43]. Therefore, we examined whether each fibroblast subpopulation upregulated genes associated with particular matrisome categories (basement membrane, interstitial collagens, ECM glycoproteins and proteoglycans; Fig. 2e). This showed that myofibroblasts upregulated multiple genes in each matrisome category, including the majority of upregulated interstitial collagens (Fig. 2e); alveolar fibroblasts

upregulated expression of multiple genes associated with basement membranes and ECM glycoproteins; and adventitial fibroblasts upregulated multiple ECM glycoproteins and proteoglycans (Fig. 2e). To examine these differences further we calculated module scores for each matrisome category and compared their overall expression between fibroblast subpopulations (Fig. 2f and Supplementary Fig. 2g). This showed that myofibroblasts significantly upregulated interstitial collagens compared to both adventitial and alveolar fibroblasts (Fig. 2f); whereas alveolar and myofibroblasts both exhibited significantly increased expression of basement membrane genes compared to adventitial fibroblasts (Fig. 2g). It is well described that excessive collagen deposition is a key role played by myofibroblasts. To determine if this was a pathology-specific function of these cells, we examined whether the expression of specific matrisome components varied within fibroblast subpopulations isolated from control or tumour tissues. This showed that interstitial collagen expression was significantly increased in tumour samples across the three subpopulations (Fig. 2h).

This result suggested there may be variation within fibroblast subpopulations driven by pathology, possibly reflecting the level of activation. To examine this further we performed differential expression analysis between control and tumour samples within each subpopulation. This identified multiple genes downregulated in tumour-associated adventitial fibroblasts compared to control counterparts, including *IGFBP6, FABP4 and DCN* (Supplementary Fig. 2g and Supplementary Data 5). In contrast, multiple genes were upregulated in tumour-associated myofibroblasts compared to their control counterparts, including *SULF1, COL11A1 and LRRC15* (Supplementary Fig. 2h and Supplementary Data 5). We also examined whether gene signatures for previously described CAF subpopulations were differentially expressed between fibroblasts from control or tumour samples. As expected, this showed that irrespective of tumour subtype, the myoCAF gene signature was significantly upregulated in tumour samples (Fig. 2i). In contrast, however, the iCAF gene signature was significantly downregulated in fibroblasts isolated from tumour samples compared to control tissue (Fig. 2j).

In summary, these data show that the fibroblasts present in NSCLC are consistent with the three major subpopulations identified as tissue-resident in non-cancerous lung tissue[18] and are likely to differentially regulate ECM maintenance/remodelling. These data also suggest that interaction with NSCLC tumours results in significant changes in gene expression within each of these subpopulations, consistently involving the upregulation of interstitial collagens in addition to subpopulation-specific changes in phenotype. Additionally, we have shown that myoCAF signatures were increased in NSCLC compared to control lung fibroblasts, whereas iCAF gene signatures were increased in control lung tissues.

## Investigating spatial distribution and abundance across subtypes

To orthogonally validate the three fibroblast subpopulations identified by scRNA-seq, we designed a multiplexed immunohistochemistry (mxIHC) panel. For this analysis, the human protein atlas[44,45] database was used to identify marker genes for each subpopulation that had both 'enhanced' antibody validation, indicating concordant expression levels when measured by IHC or RNA-seq; and documented intracellular detection of the protein (Fig. 3a). Genes passing these criteria were screened, using the human protein atlas images, by a consultant pathologist for expression in fibroblasts and CD34, AOC3 and POSTN or ACTA2 (α-SMA) were selected as optimal IHC markers for adventitial, alveolar and myofibroblasts respectively (Fig. 3a and Supplementary Fig. 3a). Additionally, exclusion markers Pan-CK, CD31 and MCAM (epithelial, endothelial and mural cell markers respectively) were incorporated into the mxIHC panel.

Consistent with previously defined nomenclature for the lung fibroblast phenotypes, our mxIHC showed the alveolar and adventitial

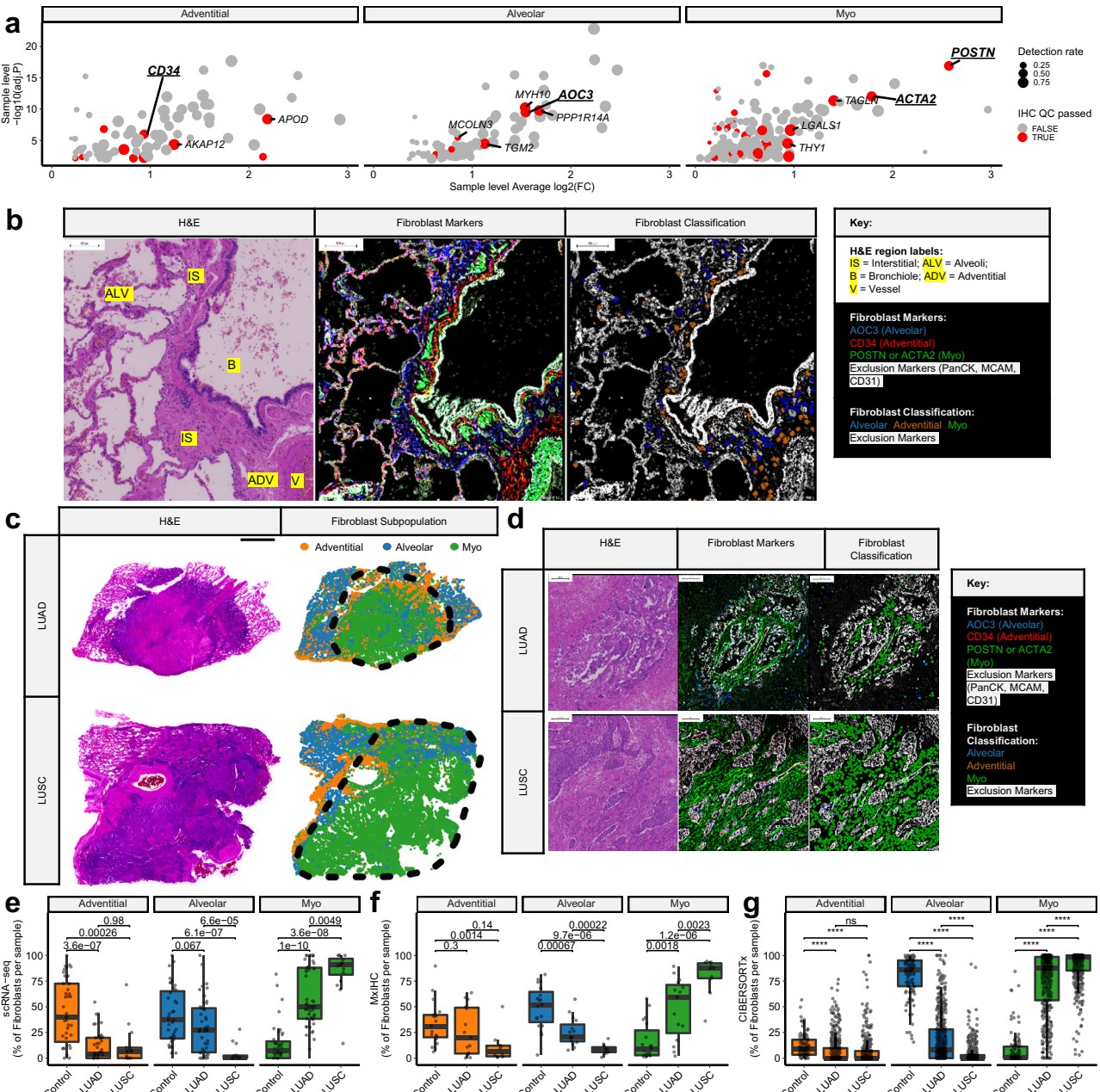

**Fig. 3 | Multiplexed IHC (mxIHC) and digital cytometry show that fibroblast subpopulations occupy spatially discrete niches and varied NSCLC tissue subtype enrichment. a** Scatter plot showing differential expression statistics (Bonferroni adj.*P* and log2 fold change) for putative subpopulation markers, highlighting those compatible for use in mxIHC analysis. Full details are provided in Supplementary Data 2. **b** Representative micrographs from H&E staining and multiplexed IHC (mxIHC) performed on serial sections (*n* = 19 samples analysed). The micrographs (from left to right) represent H&E staining (notable regions are highlighted as described in the key), a pseudocoloured image depicting different markers identified by mxIHC (coloured as indicated in the key) and the results from histo-cytometry-based cell classification (with simulated cells coloured as indicated in the key). The region presented represents histologically normal lung tissue, including adventitial, alveolar, peri-bronchial and interstitial regions. Scale bars represent 100 μm. For further images showing region of interest selection and individual marker staining profiles please see Supplementary Data 6. **c** Representative examples from whole slide mxIHC analysis of NSCLC tissue sections (*n* = 15 [LUAD], 10 [LUSC] samples analysed). Panels showing serial section H&E staining and a point pattern plot showing the spatial distribution of different fibroblast populations (measured by histo-cytometry analysis of mxIHC). The scale

bar represents 4 mm and the black dotted line demarcates the tumour region in each tissue section. For details of the full cohort analysed, please see Supplementary Data 7, 8. **d** As per **b**, presenting regions of the respective LUAD and LUSC cases shown in panel **c**. For further images showing region of interest selection and individual marker staining profiles please see Supplementary Data 9, 10 for the LUAD and LUSC case respectively. **e** Boxplot showing the relative abundance of each fibroblast subpopulation in different NSCLC subtypes, as measured in the integrated scRNA-seq dataset. Nominal *p* values for the Wilcoxon signed-ranks test are also shown (*n* = 39 [Control], 46 [LUAD], 16 [LUSC]). **f** As per **d**, measured by mxIHC, analysing pathologist annotated tumour regions for LUAD or LUSC and non-neoplastic regions within the tissue blocks as controls. Nominal *p* values for the Wilcoxon signed-ranks test are also shown (*n* = 19 [Control], 15 [LUAD], 10 [LUSC]). **g** As per **c**, as measured by CIBERSORTx-mediated digital cytometry. �⁰⁵*p* > 0.05, ****p < 2.22e-16, Wilcoxon signed-ranks test (*n* = 110 [Control], 515 [LUAD], 501 [LUSC])[24,25]. All statistical tests carried out were two-sided and boxplots are displayed using the Tukey method (centre line, median; box limits, upper and lower quartiles; whiskers, last point within a 1.5x interquartile range). Source data for panels **e**–**g** are provided in the Source Data file.

subpopulations were found within these regions of control (histologically normal) lung tissue (Fig. 3b and Supplementary Data 6). Notably, however, AOC3 + (alveolar) fibroblasts were also observed within interstitial lung tissue (Fig. 3b and Supplementary Data 6) and CD34 + (adventitial) fibroblasts were also observed in peri-bronchial regions (Fig. 3b and Supplementary Data 6).

This mxIHC panel was then applied to whole-tissue sections of NSCLC tissue (15 LUAD and 10 LUSC, including ten samples from our scRNA-seq cohort, Supplementary Data 7, 8). As expected, our three fibroblast subpopulations identified discrete subpopulations of fibroblasts (Supplementary Fig. 3b, c) and each subpopulation was found in the three tissue types analysed (control, LUAD and LUSC; Fig. 3c, d and Supplementary Data 9, 10).

To examine these fibroblast phenotypes across larger cohorts, CIBERSORTx-mediated digital cytometry was performed[27], using scRNA-seq data to generate a signature matrix: consisting of each fibroblast subpopulation, endothelial cells, mural cells, epithelial cells and immune cells (Supplementary Fig. 3d and Supplementary Data 11). The accuracy of this method was tested using a pseudobulk dataset generated by combining single-cell transcriptomes, providing ground truth for each pseudobulk sample (Supplementary Fig. 3e). This confirmed accuracy in enumerating all cell types, including the three fibroblast subpopulations ($R^2 = 0.81$ [adventitial], 0.91 [alveolar] and 0.92 [myo]; $p < 3.53\text{e-}76$; Supplementary Fig. 3e).

We examined the relative abundance of each subpopulation in control, LUAD and LUSC tissue samples. In the scRNA-seq dataset, adventitial fibroblasts were significantly more abundant in control tissue samples compared to both LUAD and LUSC (Fig. 3e). Alveolar fibroblasts were similarly most abundant in control tissues but were also detected at high levels in some LUAD samples and rarely present in LUSC (Fig. 3e). In contrast, myofibroblast abundance was increased in LUAD and LUSC compared to control tissues, but also significantly more abundant in LUSC compared to LUAD (Fig. 3e). These associations between fibroblast subpopulations and tissue type were confirmed using CIBERSORTx to analyse TCGA RNA-seq datasets[24,25] (Fig. 3g). Then further validated by mxIHC (Fig. 3f), where entrapped non-neoplastic tissue within the tissue block was excluded from the analysis (as shown in Fig. 3c).

Given that LUSC tumours were found to have higher levels of myofibroblasts than LUAD tumours, we hypothesised that tumour subtypes may also differentially impact adjacent tissues. To examine this, we used our mxIHC dataset to compare tumour-adjacent tissue regions in LUSC and LUAD cases. This showed that, similar to the intratumour regions, LUSC-adjacent areas of the lung had increased myofibroblasts and decreased alveolar fibroblasts compared to LUAD-adjacent lung tissue (Supplementary Fig. 3f). These tumour-adjacent regions were then assessed by a pathologist to determine whether there was evidence of inflammation and/or fibrosis, which was found in 55% of LUSC cases but not in LUAD (Supplementary Fig. 3g). Significant differences in the abundance of alveolar and myofibroblasts were also found when grouping these control tissue regions by the presence of inflammation and/or fibrosis (Supplementary Fig. 3h). However, no significant difference in the abundance of adventitial fibroblasts was found in either comparison.

In summary, adventitial and alveolar fibroblast subpopulations were enriched in control lung tissues and replaced by myofibroblasts in NSCLC. Furthermore, fibroblast subpopulation abundance also differs between NSCLC subtypes, with LUAD tumours exhibiting greater heterogeneity between the three subpopulations, whereas LUSC tumours have consistently high levels of myofibroblasts.

## Investigating MyoCAF activation with trajectory inference

To examine the process of transdifferentiation from alveolar and adventitial fibroblast subpopulations (enriched in control tissue) to myofibroblasts (enriched in tumour tissue) we performed trajectory inference on our scRNA-seq dataset, using a diffusion map dimensionality reduction[46]. This showed adventitial and alveolar fibroblasts may act as independent progenitors from which myofibroblasts can transdifferentiate (Fig. 4a, b). We then ordered cells in 'pseudotime', representing their relative progression towards a myofibroblast phenotype (Fig. 4c). Genes differentially expressed in pseudotime were identified by examining each dataset individually and then calculating meta $p$ values using Stouffer's method (Supplementary Data 12). Significant genes (adjusted meta-$p < 1 \times 10^{-10}$ and nominal $p < 0.05$ in at least three datasets) were then clustered into modules based on correlated expression in 'pseudotime'. This identified four modules on both the trajectories representing different phases of transdifferentiation: progenitor, early-activation, proto-differentiation and differentiation (Fig. 4d, e). Comparing the genes assigned to each of these modules highlighted significant overlap across the two trajectories, suggesting that the process of myofibroblast transdifferentiation is similar irrespective of the pre-cursor subpopulation (Fisher's exact $p < 1.03\text{e-}10$; Supplementary Fig. 4c).

To functionally annotate these phases of transdifferentiation, we performed enrichment analysis, using the REACTOME pathway database (Fig. 4h and Supplementary Data 13). The Early-activation module was significantly enriched with genes known to be involved in cytokine signalling (Fig. 4h) and also contained multiple previously described iCAF marker genes[8] (e.g. *IL6* and *CCL2*). Notably, similar expression profiles were found when analysing datasets consisting of only control samples or both control and NSCLC samples (Fig. 4f and Supplementary Fig. 4b), suggesting that this phase of the transdifferentiation process may be independent of stimuli associated with NSCLC tumours. To confirm this, we grouped all fibroblasts in the scRNA-seq dataset into five bins based on their position in pseudotime and then calculated sample averages for the expression of each module. This showed that the Early-activation module was significantly increased in fibroblasts from control samples compared to tumour samples at all stages of the transdifferentiation process (Fig. 4g).

In contrast, expression trends over pseudotime for the proto-differentiation and differentiation modules were qualitatively different between datasets consisting of only control samples or control and NSCLC samples (Fig. 4f and Supplementary Fig. 4b). Therefore, we applied the same approach to determine whether sample type impacted the expression of the Proto-differentiation module. This identified a significant increase in tumour samples at the intermediate stage of transdifferentiation (Fig. 4g). This module was enriched with many genes that encode ribosomal proteins (Fig. 4h), which is a well-described ultrastructural feature of myofibroblasts and may indicate that these cells have increased capacity to perform protein translation[47]. Heat-shock family genes involved in HSF1 transactivation (*e.g. HSPA1A, DNAJB1* and *HSP9OAB1*) were also significantly enriched in this module (Fig. 4h), which has been shown previously to regulate cancer-mediated fibroblast activation[48]. Furthermore, genes involved in oxidative stress responses (e.g. *HIF1A, GGT* and *SERPINE1*), which is a well-described driver of both heat shock response and fibroblast activation, were also found in this module. To investigate the role of heat shock/stress responses in myofibroblast activation further, we used mxIHC to examine HSPA1A/Hsp70 expression. This identified a significant increase in each of the fibroblast subpopulations when comparing cells that were located within the tumour to those in tumour-adjacent control regions (Fig. 4i), confirming that stimuli from NSCLC tumours induce the heat shock response to a greater extent than stimuli from control tissues.

The Differentiation module was significantly increased in tumour tissues compared to control at all stages of transdifferentiation (Fig. 4g). This module was enriched with genes involved in collagen formation and ECM organisation (Fig. 4h), consistent with the myofibroblast phenotype and data described above showing that fibrillar collagens are upregulated in tumour samples compared to control tissue samples.

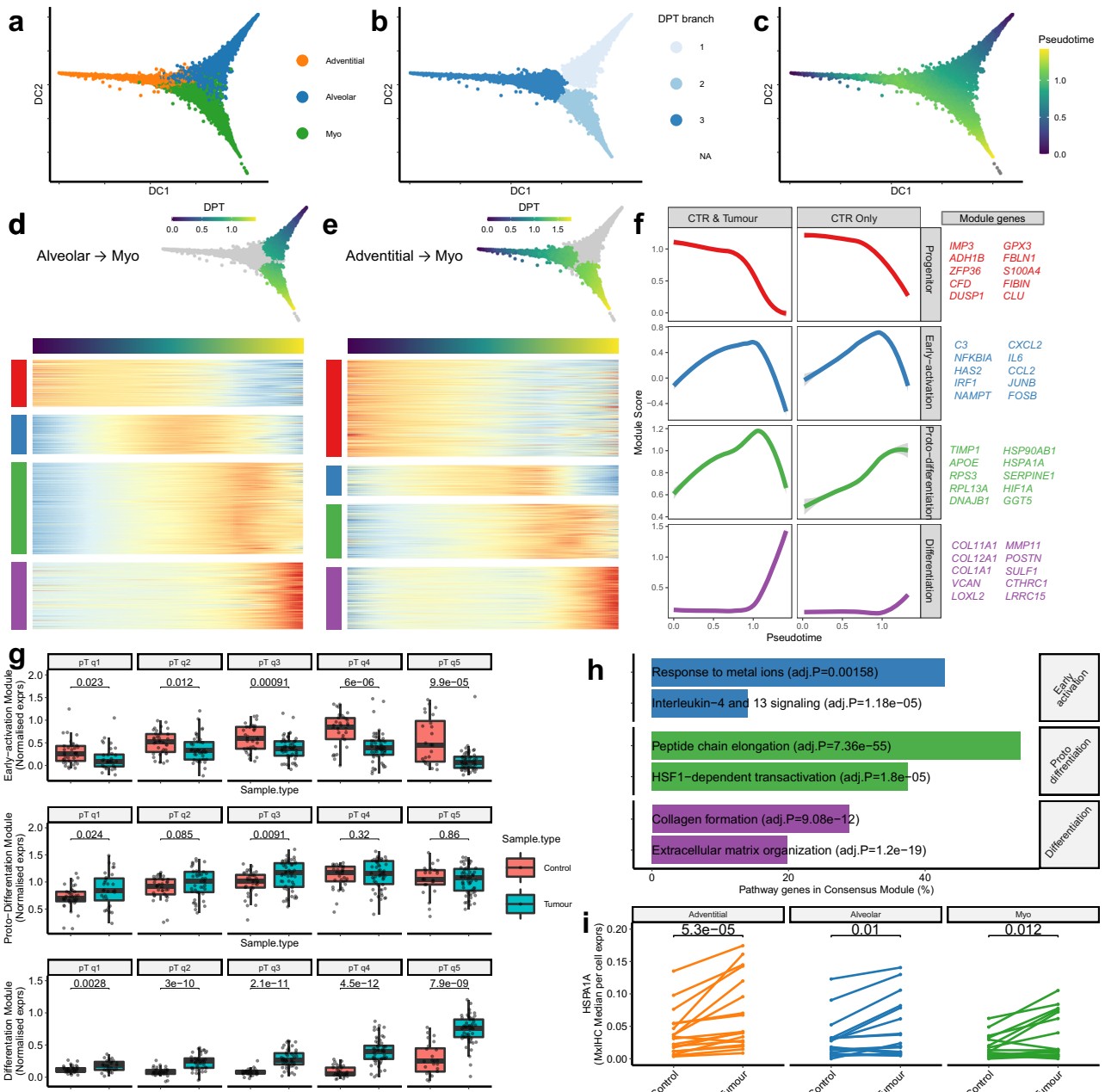

**Fig. 4 | Trajectory inference identifies consensus gene modules associated with the transdifferentiation from alveolar or adventitial fibroblasts to myofibroblasts. a** 2D visualisation (diffusion map dimensionality reduction) of the integrated fibroblast scRNA-seq dataset, highlighting the three subpopulations. **b** As per **a**, highlighting cells assigned to each diffusion pseudotime (DPT) branch. **c** As per **a**, showing the relative position of each cell in DPT. **d** Heatmap showing genes differentially expressed as alveolar fibroblasts progress to myofibroblasts in DPT. Hierarchical clustering was used to group these genes into modules defined by the DPT expression profile. Complete differential expression results are provided in Supplementary Data 12. **e** As per **d**, genes differentially expressed as adventitial fibroblasts progress to myofibroblasts. Complete differential expression results are provided in Supplementary Data 12. **f** Loss curve plots showing the expression profiles for each consensus DPT module in datasets consisting of control and tumour samples or only control tissue samples. Consensus modules consist of genes assigned to the same cluster in both alveolar to myo and adventitial to myo trajectories, ten representative genes for each module are listed (full gene lists can be found in Supplementary Data 12 and individual dataset plots are shown in Supplementary Fig. 4b). **g** Boxplots showing the expression of consensus DPT modules in cells grouped by tissue type and pseudotime quintiles. Nominal *p* values for the Wilcoxon signed-ranks test are also shown (*n* Control/Tumour = 37/32 [q1], 36/45 [q2], 36/55 [q3], 34/58 [q4], 24/58 [q5]). **h** Barplots showing REACTOME pathway enrichment results for each consensus DPT module (Benjamini−Hochberg adjusted *P* values are also shown). Full results provided in Supplementary Data 13. **i** Line plots showing the average HSPA1A expression levels between paired tumour and adjacent normal tissues for each fibroblast subpopulation, measured by mxIHC. Wilcoxon signed-ranks test (*n* = 18). All statistical tests carried out were two-sided and boxplots are displayed using the Tukey method (centre line, median; box limits, upper and lower quartiles; whiskers, last point within a 1.5x interquartile range). Source data for panels **g**, **i** are provided in the Source Data file.

In summary, these results suggest that both control tissue-enriched fibroblast subpopulations (adventitial and alveolar) could act as tissue-resident progenitors for myofibroblasts. These data also suggest that the process of transdifferentiation is comparable irrespective of the progenitor subpopulation: involving a transient phase of inflammatory gene upregulation, independent of direct interaction with the tumour; followed by proto-differentiation involving heat shock response signalling, which is increased by interaction

with the tumour; finally leading to a fully differentiated myofibroblast phenotype with increased capacity for ECM organisation and collagen formation, which is significantly enhanced by tumour-associated stimuli.

## Cross-tissue analysis of fibroblast phenotypes

To examine whether these fibroblast phenotypes were conserved across cancer types, we analysed publicly available data from PDAC[49], HNSCC[29] and colorectal cancer (CRC)[50]. In each case, fibroblasts were identified by unsupervised clustering followed by mural cell exclusion, as described above (Fig. 5a). Then probabilistic machine learning models were used to classify these cells as one of the three sub-populations identified in NSCLC (Fig. 5b, c). This showed that the adventitial and myofibroblast populations were strongly conserved across all cancer types analysed (adventitial median probability = 0.67 [PDAC], 0.75 [HNSCC], 1 [CRC]; myo median probability = 1 [PDAC], 1 [HNSCC], 1 [CRC]); whereas fibroblasts assigned to the alveolar sub-population had consistently lower probability scores (median probability = 0.49 [PDAC], 0.52 [HNSCC] and 0.56 [CRC]), suggesting a greater degree of phenotypic divergence from the lung.

We then validated these results by applying our multiplex IHC panel to tissue microarrays consisting of tumour and control tissue cores from PDAC, HNSCC and CRC. Consistent with the scRNA-seq findings, this showed that adventitial and myofibroblasts were the predominant subpopulations identified in each of these cancer types (Fig. 5d and Supplementary Fig. 5a). Furthermore, as we found in NSCLC, adventitial fibroblasts were significantly more abundant in control compared to tumour tissues across all three tumour types (Fig. 5e) and myofibroblasts more abundant in tumour tissues (Fig. 5f).

To test whether the alveolar phenotype was specific to lung tissues, we performed a similar analysis on scRNA-seq data generated from idiopathic lung fibrosis (IPF) samples[15] (i.e. a non-cancerous lung pathology). This showed that all three subpopulations were identified with high probability scores (median probability = 0.95 [Adventitial], 0.88 [Alveolar], 0.90 [Myo]; Supplementary Fig. 5b–d Notably, this analysis also showed that probability associated with myofibroblast classifications was lower for IPF than in the cancer datasets (Supplementary Fig. 5e), suggesting there may be subtle differences between myofibroblasts found in cancer and fibrosis.

## Survival analysis using multiple NSCLC cohorts

To examine the clinical relevance of fibroblast subpopulations in NSCLC, we leveraged imputed cell abundances from CIBERSORTx-mediated digital cytometry to interrogate large clinically annotated NSCLC cohorts (four LUAD cohorts, $n = 1669$;[25,51–53] four LUSC cohorts, $n = 1104$;[24,54,55] Supplementary Data 14). The relative abundance of each fibroblast subpopulation was used in Cox proportional hazards regression modelling. This identified a consistent link between myo-fibroblasts and poor overall survival in LUAD ($p < 0.01$; Supplementary Fig. 6a), but no significant correlation was found in LUSC (Supplementary Fig. 6a). Given that myofibroblasts are highly abundant in both NSCLC subtypes we hypothesised that this difference could be due to phenotypic changes between subtypes. However, no genes were identified as significantly differentially expressed through sample-level analysis, suggesting minimal phenotypic variance at the transcriptome level between myofibroblasts from these two NSCLC subtypes (Supplementary Fig. 6b).

To examine the potential for using myofibroblast abundance as a prognostic biomarker for patient stratification in LUAD, we used TCGA-LUAD dataset as a test cohort to determine the optimal threshold for categorising samples as myofibroblast high (>85.2%) and low (<85.2%; Fig. 6a–c). We then applied this threshold to three vali-dation cohorts, demonstrating consistently significant patient strati-fication (log-rank $p \leq 0.02$; Fig. 6d). Multivariate cox regression analysis also showed that these prognostic correlations were

independent of disease stage and patient age ($p < 0.0001$, HR [95% CIs] = 1.70 [1.38, 2.09]; Fig. 6i and Supplementary Fig. 6c).

In contrast, alveolar and adventitial fibroblast abundance was associated with better overall survival rates in multiple LUAD datasets (Supplementary Fig. 6a). This association was particularly consistent for alveolar fibroblasts, which were significant across all datasets analysed ($p < 0.01$ and Supplementary Fig. 6a). Therefore, we applied the same approach as described above to test the potential for using alveolar fibroblast abundance as a prognostic marker (Fig. 6e–g). Similarly, this showed that dichotomising LUAD cohorts as alveolar fibroblast high (>22.0%) or low (<22.0%) was consistently effective at stratifying overall survival rates (Log-rank $p < = 0.02$; Fig. 6h); and that this association was independent of disease stage and patient age (Fig. 6j and Supplementary Fig. 6d).

## Investigating associations with key prognostic features of LUAD

The morphological subtype of LUAD tumours is recognised to be associated with patient survival rates[56]. Fibroblast subpopulation abundance significantly varied between morphological subtypes of LUAD, as shown by CIBERSORTx ($n = 623$, $p < 0.0001$; Fig. 7b), scRNA-seq ($n = 21$; Fig. 7c) and mxIHC ($n = 15$; Fig. 7a, d). Myofibroblasts were increased in poorly differentiated (G3; solid or micropapillary) tumours compared to moderate/well-differentiated (G1/G2; lepidic, acinar and papillary). Despite this association, fibroblast subpopulation abundance remained a significant independent prognostic indicator in multivariate Cox regression, including age, stage and grade as covariates (myofi-broblasts: HR [95% CIs] = 1.44 [1.07, 1.95], adj.$P = 0.015$; alveolar fibro-blasts: HR [95% CIs] = 0.67 [0.46,0.96], adj.$P = 0.028$; $n = 601$).

LUAD morphology can be heterogeneous within individual tumours. Therefore, we used our MxIHC dataset to examine the association between fibroblast subpopulations and specific morpho-logical patterns (Supplementary Fig. 7b). This identified a significant correlation between myofibroblasts and the percentage of the tumour comprised of solid growth patterns (rho = 0.60, $p < 0.01$; Supplemen-tary Fig. 7c); and myofibroblasts were clearly observed to be the principal stromal cell type found in solid regions of mixed morphology tumours (Supplementary Data 15). A weaker and non-significant cor-relation (rho = 0.44, $p = 0.07$) was observed with micropapillary growth patterns (Supplementary Fig. 7b).

Previous studies have described a link between the morphological and molecular subtypes (proximal-inflammatory [PI], proximal-proliferative [PP] and terminal respiratory unit [TRU]) of LUAD[25]. We confirmed this in the bulk tissue datasets, finding that 77% of TRU tumours were moderate/well-differentiated (G1/G2) and 69% of PP tumours were poorly differentiated (G3). As expected, given this link, myofibroblast abundance was highest in PP tumours; whereas alveolar and adventitial fibroblasts were most prominent in TRU tumours (Fig. 7f and Supplementary Fig. 7d). Furthermore, consistent with previously described associations between PP tumours and *TP53* mutations[23], we also found that myofibroblast abundance was increased in LUAD tumours harbouring *TP53* mutations (Fig. 7e and Supplementary Fig. 7e).

We also used CIBERSORTx to examine the abundance of immune cell subpopulations (LM22[57]) and their correlation with fibroblast subpopulations. This demonstrated an inverse relationship between the immune cells that correlated with myofibroblasts and alveolar fibroblasts, which was consistently observed across all the LUAD transcriptomic datasets analysed (Fig. 7g and Supplementary Fig. 7f). Showing that alveolar fibroblasts were correlated to multiple resting immune cell subsets (Fig. 7g and Supplementary Fig. 7f; e.g. mast cells, CD4 + memory T-cells and dendritic cells), in addition to monocytes and B cells (both memory and naive subsets). In contrast, myofibro-blasts correlated with macrophages, neutrophils, activated mast cells and activated CD4 + memory T-cells (Fig. 7g and Supplementary Fig. 7f). Indicating that myofibroblast differentiation within the tumour

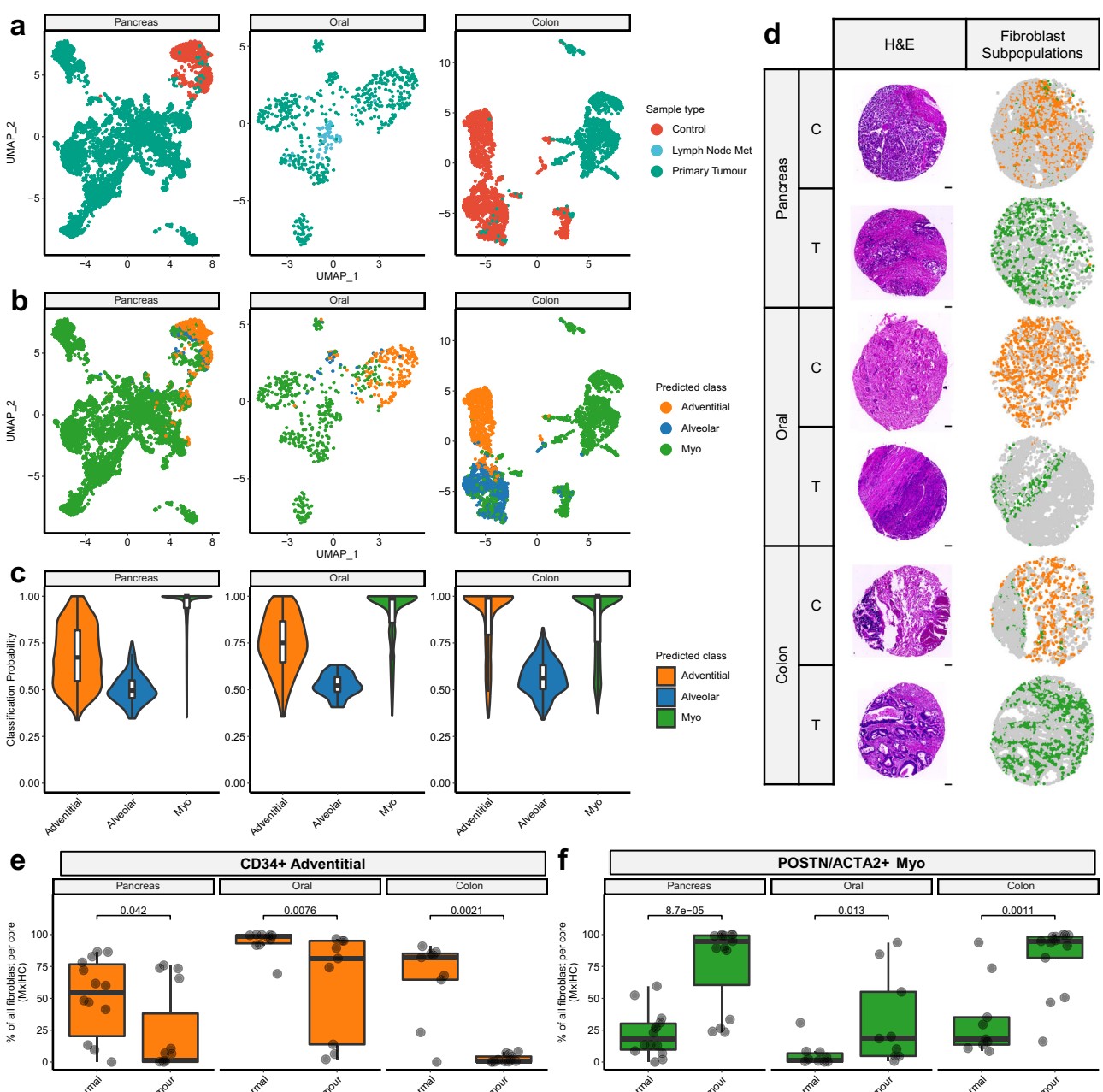

**Fig. 5 | Machine learning-based classification of scRNA-seq data and mxIHC shows adventitial and myofibroblasts are conserved across pancreatic, colorectal and oral cancers, whereas alveolar fibroblasts are lung-specific. a** 2D visualisation (UMAP dimensionality reduction) of fibroblasts isolated from different cancer types and analysed by scRNA-seq, highlighting the sample type. **b** As per **a**, highlighting the fibroblast subpopulation associated with each cell as predicted by a machine learning classifier. **c** Violin plots showing the probability of the machine learning classifier model's predictions grouped by subpopulation (*n* = 6875 [Pancreas], 611 [Oral] and 3297 [Colon] fibroblast transcriptomes). **d** Representative images from mxIHC analysis of tissue microarrays (TMAs), constructed from pancreatic, oral and colon cancer tissue blocks (n cores analysed Control/Tumour = 14/15 [Pancreas], 10/9 [Oral], 9/13 [Colon]). Showing H&E

staining and the spatial distribution of fibroblast subpopulations (measured by histo-cytometry analysis of mxIHC data). The scale bar represents 100 μm. Further images of individual staining profiles are provided in Supplementary Fig. 5a. **e** Boxplot showing the relative abundance of adventitial fibroblasts in tumour or control tissues, measured by mxIHC analysis of TMA cores. Nominal *p* values for the Wilcoxon signed-ranks test are also shown (n Control/Tumour = 14/15 [Pancreas], 10/9 [Oral], 9/13 [Colon]). **f** As per **e**, for myofibroblast abundance. All statistical tests carried out were two-sided and boxplots are displayed using the Tukey method (centre line, median; box limits, upper and lower quartiles; whiskers, last point within a 1.5x interquartile range). Source data for panels **e**, **f** are provided in the Source Data file.

microenvironment is also associated with the activation/differentiation of helper T-cells and myeloid cells.

## Discussion

We have performed a comprehensive analysis of the fibroblast landscape in human NSCLC, identifying three major subpopulations present in both control and tumour tissues: adventitial, alveolar and myofibroblasts. We also show that cancer-associated myofibroblasts could originate from tissue-resident adventitial or alveolar fibroblasts through a process involving transient phases of inflammatory and stress-response signalling. Fibroblast subpopulation abundance varied between control and tumour tissues and across NSCLC histological

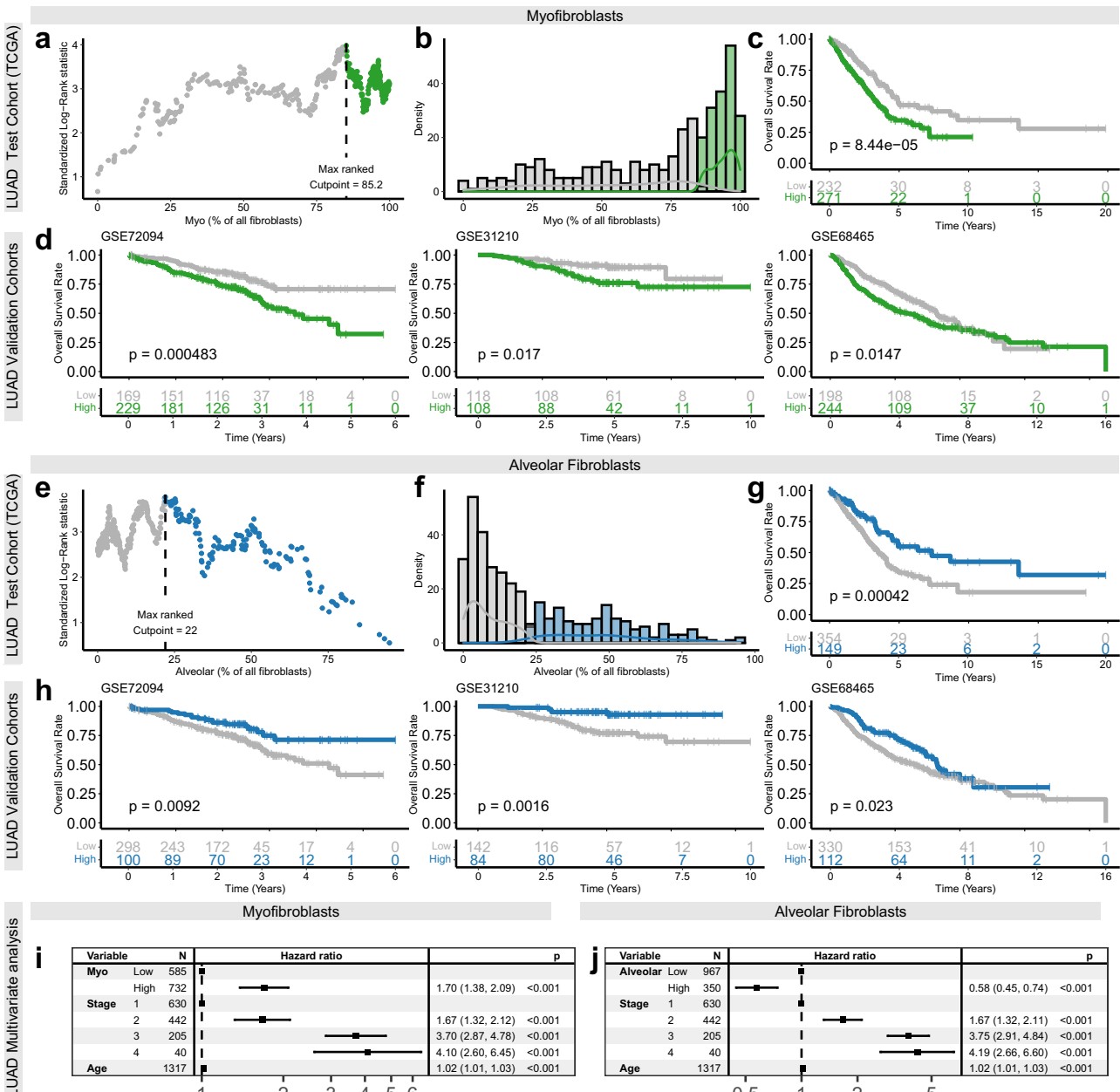

**Fig. 6 | CIBERSORTx-mediated digital cytometry shows that myofibroblasts and alveolar fibroblasts correlate with overall survival rates in LUAD. a** Scatter plot showing the variation in standardised log-rank survival statistics (correlation with overall survival rates) using different cut-points for dichotomising TCGA-LUAD[25] samples by myofibroblast abundance. **b** Density plot showing the distribution of myofibroblast abundance measurements across TCGA-LUAD[25] samples. **c** Kaplan–Meier plot showing TCGA-LUAD[25] cohort patient survival rates, stratified by myofibroblast abundance using the optimal cut-point for dichotomisation (identified in **a**). Statistical significance was assessed using a log-rank test (*n* = 503). **d** Kaplan–Meier plots showing the cutpoint defined on TCGA-LUAD cohort (identified in panel **a**) applied to three additional LUAD validation cohorts. Statistical significance was assessed using Log-rank tests (*n* = 398 [GSE72094[51]], 226 [GSE31210[52]], 422 [GSE68465[53]]). **e** Scatter plot showing the variation in standardised log-rank survival statistic (correlation with overall survival rates) using different cut-points for dichotomising TCGA-LUAD[25] samples by alveolar fibroblast abundance. **f** Density plot showing the distribution of alveolar fibroblast abundance measurements across TCGA-LUAD[25] samples. **g** Kaplan–Meier plot showing TCGA-

LUAD[25] cohort patient survival rates, stratified by alveolar fibroblast abundance using the optimal cut-point for dichotomisation (identified in **e**). Statistical significance was assessed using a Log-rank test (*n* = 503). **h** Kaplan–Meier plots showing the cutpoint defined on TCGA-LUAD cohort (identified in panel **a**) applied to three additional LUAD validation cohorts. Statistical significance was assessed using Log-rank tests (*n* = 398 [GSE72094[51]], 226 [GSE31210[52]], 422 [GSE68465[53]]). **i** Forest plot showing covariate independent hazard ratios (±95% confidence intervals) and adjusted *p* values from multivariate Cox regression analysis of four-year overall survival rates across all LUAD patient cohorts analysed above, using myofibroblast abundance, disease stage and patient age as independent variables (exact *p* values provided in Source Data file). Results for individual datasets are shown in Supplementary Fig. 6c. **j** Forest plot showing covariate independent hazard ratios (±95% confidence intervals) and adjusted *p* values from multivariate Cox regression analysis of 4-year overall survival rates across all LUAD patient cohorts analysed above, using alveolar abundance, disease stage and patient age as independent variables (exact *p* values provided in Source Data file). Results for individual datasets are shown in Supplementary Fig. 6d.

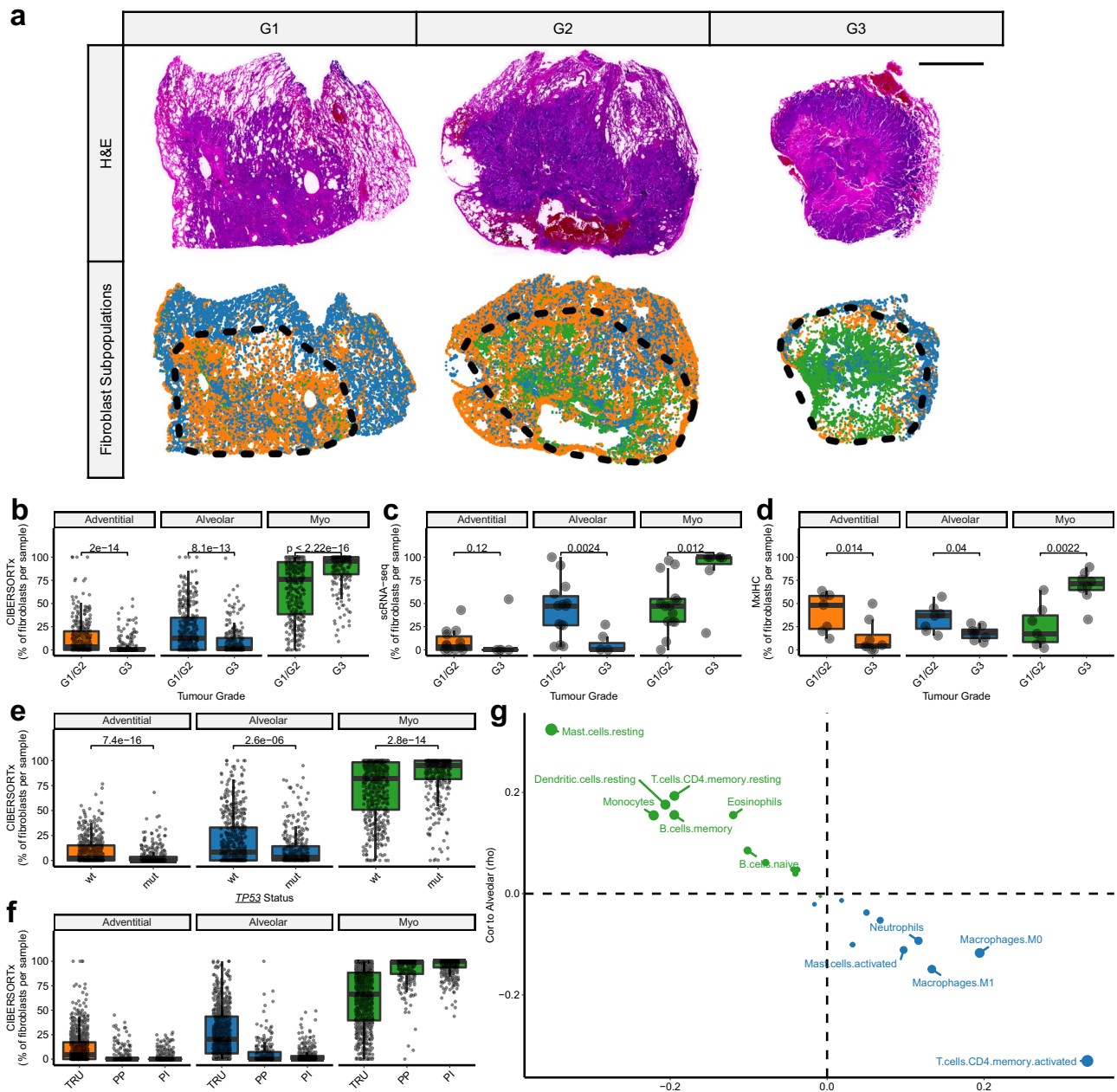

**Fig. 7 | MxIHC and digital cytometry shows fibroblast subpopulations are associated with morphological, molecular and immunological features of LUAD tumours. a** Representative images from whole slide mxIHC analysis of LUAD tumours with varying grades (G1 = well differentiated [*n* = 1], G2 = moderately differentiated [*n* = 6] and G3 = poorly differentiated [*n* = 8]), showing H&E staining and point pattern plots representing the spatial distribution of different fibroblast subpopulations (measured by histo-cytometry analysis of mxIHC data). The scale bar represents 4 mm and the black dotted line demarcates the tumour region in each tissue section. **b** Boxplots showing the relative abundance of fibroblast subpopulations in well/moderately differentiated LUAD tumours compared to poorly differentiated, measured by CIBERSORTx digital cytometry. Wilcoxon signed-ranks test (*n* = 375 [G1/G2], 248 [G3]; from two independent datasets;[25,53] data from each individual dataset is shown in Supplementary Fig. 7a). **c** As per **b**, measured by scRNA-seq. Wilcoxon signed-ranks test (*n* = 14 [G1/G2], 7 [G3]). **d** As per **b**, measured by mxIHC. Wilcoxon signed-ranks test (*n* = 7 [G1/G2], 8 [G3]). **e** Boxplots showing the relative abundance of fibroblast subpopulations in LUAD tumours grouped by

*TP53* mutation status, measured by CIBERSORTx digital cytometry. Wilcoxon signed-ranks test (*n* = 578 [wt], 374 [mut]; from two independent datasets[25,51], data from each individual dataset is shown in Supplementary Fig. 7d). **f** Boxplots showing the relative abundance of fibroblast subpopulations in LUAD tumours grouped by molecular subtype (TRU terminal respiratory unit, PP proximal-proliferative and PI proximal-inflammatory). Each datapoint represents an individual patient, measured by CIBERSORTx digital cytometry (*n* = 1626, from four independent datasets;[25,51–53] data from each individual dataset is shown in Supplementary Fig. 7c). **g** Scatter plot showing Spearman's correlation between alveolar or myofibroblast abundance and immune cell subpopulation (LM22) abundance, measured by CIBERSORTx digital cytometry (*n* = 1626; from four independent datasets;[25,51–53] data from each individual dataset is shown in Supplementary Fig. 7e). All statistical tests carried out were two-sided and boxplots are displayed using the Tukey method (centre line, median; box limits, upper and lower quartiles; whiskers, last point within a 1.5x interquartile range). Source data for panels **b**–**f** are provided in the Source Data file.

subtypes, providing significant prognostic value in LUAD, where they were associated with molecular, morphological and immune features of these tumours.

Fibroblast heterogeneity is an emerging area of cancer research with previously described relevance to patient outcomes and treatment response across multiple tumour types[7,58]. In this field, consensus terminology for different subpopulations is yet to be fully defined. Here, we demonstrate that understanding both tissue-resident fibroblasts and their cancer-associated counterparts provides a framework for establishing generally applicable terminology. The two fibroblast subpopulations more commonly found in control tissue samples (alveolar and adventitial fibroblasts) were associated with improved survival rates in LUAD, particularly alveolar fibroblasts. These findings support the rationale for developing therapeutic strategies that revert CAFs to a 'normal' phenotype, as described for vitamin D receptor agonists[59] and NOX4 inhibition[1,60]. Our cross-tissue analysis showed that alveolar fibroblasts were relatively specific to lung tissues (compared to the adventitial and myofibroblast populations). However, this does not preclude the existence of similarly tumour-suppressive tissue-resident fibroblast populations in other organs. For example, a broad analysis of murine tissues has previously identified organ-specific fibroblast subsets in the intestine, as well as bone, lung, lymph node/spleen and artery/tendon[19].

The adventitial fibroblast subpopulation is consistent with multiple previous studies[18,19] and may represent 'universal' fibroblasts[19]. We identified CD34 as an effective marker of this population, which is consistent with many studies that have described CD34 + fibroblast/stromal cell populations across different organs. For example, PDGFRA + CD34 + cells represent >90% of stromal cells within the dermis;[61] they have been shown to represent a key progenitor cell in tissue regeneration[62] and a prominent source of tumour-reactive stromal cells[63]. The adventitial fibroblasts were also the closest 'relative' to iCAF phenotypes[8]. However, in contrast to previous studies, we found that the expression of inflammation-associated cytokines and iCAF markers were predominantly limited to control tissues in NSCLC. Furthermore, our trajectory analysis suggests this inflammatory phenotype may represent an early and transient phase in the transdifferentiation to a myofibroblast phenotype, independent of direct interaction with the tumour. Indeed, inflammatory fibroblasts have been identified in infections and inflammatory conditions as well as cancer;[64] supporting the concept that fibroblasts progress through a spectrum of phenotypes that function to initiate, support and ultimately suppress inflammation.

Previous studies have shown that fibroblast markers can indicate both good and bad prognosis in NSCLC[65,66]. Our results suggest these inconsistencies are likely due to both fibroblast heterogeneity and variation across NSCLC histological subtypes. This may be because LUSC (compared to LUAD) have a more homogeneous stroma containing consistently high levels of myofibroblasts, which could limit their utility as a prognostic marker. Alternatively, the functional impact of myofibroblasts on tumour progression may vary between tumour subtypes. Multiple mechanisms for how these cells could promote tumour progression have been identified in pre-clinical cancer models: including modulating ECM organisation and tumour cell invasion[67]; promoting tumour cell proliferation[68]; regulating epithelial differentiation[69] and promoting immune evasion[7,60,70]. However, recent studies have also demonstrated that myofibroblasts can have tumour-suppressive properties in PDAC[4–6]. The observation that myofibroblasts are not prognostic in LUSC provides further evidence that the role played by these cells in cancer may be context-dependent, and further investigation is required to determine what mechanisms underlie this difference. Given that we found no significant differences in gene expression between myofibroblasts from LUAD and LUSC tumours, the varied impact these cells have on tumour progression may be due to intrinsic properties of the

malignant cells, as described in a recent study elucidating a LUAD-specific role for GREM1-KDR signalling in disease progression[68].

In summary, this comprehensive analysis of lung and NSCLC fibroblast heterogeneity shows how certain subpopulations are organ-specific, whilst others are consistently found in different tissues, cancers and pathologies. Myofibroblasts and alveolar fibroblasts are associated with different molecular and immunological LUAD subtypes, and their opposing effects on prognosis accurately identify high-risk patients. These findings could improve patient stratification and should refine strategies for therapeutic targeting of fibroblasts in lung cancer.

## Methods

### Sample acquisition and processing

The Southampton and South West Hampshire Research Ethics Committee approved the study, and written informed consent was obtained from all subjects (Target Lung study: REC number 14/SC/0186). Newly diagnosed, untreated patients with respiratory malignancies were prospectively recruited once referred. Freshly resected tumour tissue and, where available, matched control tissue was obtained from lung cancer patients following surgical resection. Control tissues were sampled from peripheral regions of the resected tissue at the furthermost point from the tumour. Tissue samples were transported (within 1 h) to the laboratory on ice in serum-free Dulbecco's Modified Eagle Medium (DMEM; Sigma-Aldrich).

Tissue disaggregation was performed as previously described[28]. Briefly, samples were washed, incised and incubated with Collagenase P (3 U/ml; Sigma) at 37 °C with agitation (200 rpm) for 60 min. The resulting suspension was strained; incubated with red cell lysis buffer (BioLegend); and re-suspended in PBS supplemented with 9% Optiprep (Sigma) and 0.1% bovine serum albumin (BSA). Single-cell transcriptome encapsulation was performed using a custom microfluidic platform (Drop-seq) as described previously[28,71].

### Single-cell RNA-seq data processing and analysis

**'Target lung drop-seq' (TLDS) dataset processing.** ScRNA-seq data processing and analysis was performed using the Seurat package in R (v4.0.2)[36], unless otherwise stated. Initial quality control was carried out to remove low-quality events. First, we used a random forest classifier to exclude empty droplets as described previously[28,72]. We then identified outliers for the fraction of reads mapping to mitochondrial genes (>2 median absolute deviations; MADs) to exclude apoptotic cells.

Initial clustering was performed using a subset of genes selected based on variance and average non-zero expression, excluding extreme outliers. Raw counts data was log-normalised and scaled (regressing out nUMI) before performing principal components analysis (PCA). Clusters were identified with the *FindClusters* function, using principal components identified as significant with *JackStraw* analysis ($p$ < 1e-5) and a resolution of 0.2. Cluster markers were identified using the *FindAllMarkers* function (ROC classifier) and cell types were assigned based on canonical marker expression or by significant enrichment (adj. $p$ < 0.0001) in previously described cell type markers from the Immunological Genome[73] and LungGENS[74] projects, assessed using the ToppFun gene set enrichment tool[75].

This identified a large cluster of cells predominantly comprised of lymphocytes. However, due to the relatively low number of genes detected in lymphocytes (described previously[12]) and, therefore, high susceptibility to false negatives in marker detection (due to drop-out), this cluster had very few genes identified as markers. Therefore, it was unclear whether these lymphocytes were sufficiently separated from low-quality droplets. To identify lymphocytes within this cluster the *AddModuleScore* function was used to calculate the average expression of previously described T-cell markers (*TRBC2, CD3D, CD3E, CD3G, CD2, IL7R* and *CD8A*) and NK cell markers (*FGFBP2, SPON2, KLRF1,*

*NKG7, PRF1* and *KLRD1*). This cluster was then filtered further to remove cells negative for both gene signatures.

**Fibroblast identification in scRNA-seq datasets.** Processed scRNA-seq datasets and associated metadata were downloaded from publicly available sources (Our data, refs. 16, 32, 18, 34, 31) for use in this study. For consistency, the cells analysed were restricted to those obtained from primary tumour tissue or control lung tissue (from NSCLC patients). Each dataset was processed using the Seurat package in R (v4.0.2) to perform log-normalisation, find variable features, principal components analysis (PCA) based dimensionality reduction and nearest neighbour graph construction[36].

Batch effects were assessed by comparing k-nearest neighbour overlap between potentially confounding sample groups and randomly sampled cells to calculate *z*-scores (nOverlap − x̄[Random samples overlap]/ σ[Random samples overlap]). This identified significant (median *z*-score >1.96) patient-dependent batch effects in each dataset. The previously described reciprocal PCA (rPCA)[35] approach for data integration was then used for intra-dataset batch correction, as implemented in the Seurat R package.

Dimensionality reduction (PCA and UMAP) and clustering was performed using the integrated (batch-corrected) data. The stromal cell cluster was then identified from canonical marker expression (e.g. *DCN, LUM* and *DPT*) and a subset from the entire dataset. The *AddModuleScore* function was then used to calculate per-cell expression levels for fibroblast and mural cell gene signatures (Fig. 1e) and fibroblasts were identified as those cells where the fibroblast signature score - mural cell signature score >0.1 and the fibroblast signature score >0.

A consensus list of fibroblast genes (*n* = 2805) was then identified as those expressed in >1% of fibroblasts in each dataset. The Seurat package *IntegrateData* function was then used to calculate rPCA corrected expression values for these consensus fibroblast genes within each dataset.

**Fibroblast meta-analysis.** The fibroblasts isolated from each scRNA-seq datasets were integrated using canonical correlation analysis (CCA) dimensionality reduction to create a shared low-dimensional space across datasets. 1422 genes (listed in Supplementary Data 16) with the highest standardised variance across all datasets were identified for this analysis. Mutual nearest neighbours (anchors) were then calculated using 30 canonical correlation vectors, filtering out potentially incorrect anchor pairs based on gene expression profiles and anchor weighting. To build the integrated dataset, anchors between all pairs of datasets were scored and then progressively merged. An integrated (batch-corrected) expression value for each of the 1422 variable genes was then calculated and used for dimensionality reduction (PCA and UMAP), nearest neighbour graph construction and clustering, as implemented in the Seurat (v4.0.2) R package. Initial clustering identified small clusters of cells marked by immune cell markers, likely to represent fibroblast/immune cell doublets or contaminating immune cells, which were excluded from further analysis.

Marker genes were identified using the *FindAllMarkers* function to apply a Wilcoxon signed-ranks test, which was applied both to the single-cell data and to sample-level data generated using the *AverageExpression* function. GSVA was performed using the GSVA (v1.36.3) package to calculate an enrichment score for each gene set per sample as the normalised difference in empirical cumulative distribution functions (CDFs) of ranked gene expression inside and outside the gene set. The limma (v3.44.3) R package was then used to assess whether these enrichment scores were differentially expressed between sub-populations using linear models.

To infer differentiation trajectories, we used the destiny (v3.2.0) R package to perform diffusion map dimensionality reduction, cluster cells by diffusion pseudotime branches and calculate pseudotime values. A loess regression model was then used to identify genes that

were differentially expressed in and calculate fitted values for each gene's expression profile in pseudotime for each dataset, using the gam (v1.20) R package. Meta-*p* values were then calculated to determine those genes that significantly varied across all datasets analysed, using the meta p (v1.5) R package. To identify pseudotime modules, the median was calculated (across datasets) for each gene's loess-fitted expression profile in pseudotime and the correlation between genes was calculated as Pearson's correlation coefficient (r). Gene modules were then identified through unsupervised clustering using "Ward's method", applied to a distance matrix constructed by calculating 1-r.

To compare fibroblast phenotypes across pathologies we analysed publicly available data from IPF[15], PDAC[49], CRC[50] and HNSCC[29]. These datasets were processed to extract fibroblasts, as described above. Integration with the NSCLC dataset was then performed by CCA, as described above. Label transfer was then performed as described previously[35], generating a predicted cluster classification and associated prediction score for each class.

## Multiplex immunohistochemistry (MxIHC)

**Staining and image capture.** Immunohistochemical staining was performed using a previously-described multiplexed protocol[76]. Four-micrometre sections of formalin-fixed paraffin-embedded (FFPE) sections were mounted on TOMO slides (Matsunami) and baked for 60 min at 60 °C. Deparaffinisation, rehydration, antigen retrieval and immunohistochemical staining were performed using the PT Link Autostainer (Dako).

Antigen retrieval for all antibodies was performed using the EnVision FLEX Target Retrieval Solution (Dako, pH indicated in Table 1). Sections were then incubated with primary antibody (details provided in Table 1) for 20 min (except for ACTA2 [αSMA], which was incubated for 10 min). Endogenous peroxidase activity was blocked using the Envision FLEX Peroxidase-Blocking reagent (Dako). EnVision FLEX HRP detection reagent (Dako) for secondary amplification and enzymatic conjugation. Chromogenic visualisation was performed using haematoxylin counterstaining and 2 × 5-min washes in either diaminobenzidine (DAB, for CD31 staining) or 3-amino-9-ethylcarbazole (AEC, for all other markers). Initially, sections were stained for CD31, then sequentially stained for additional markers. Between each staining iteration, AEC staining was removed using organic solvents (70% ethanol, 2 min; 99% ethanol, 2 min; xylene, 2 min; 99% ethanol, 2 min; 70% ethanol, 2 min); and antigen retrieval was performed in preparation for the next primary antibody and to denature antibodies from the previous staining iteration. For each staining iteration, whole slide images (WSIs) were captured at 20x with a ZEISS Axio Scan.Z1, using ZEN 2 software (ZEISS).

**Multiplex sequence optimisation and validation.** The method used for mxIHC repeatedly exposes tissue sections to organic solvents to strip the AEC stain and the EnVision FLEX Target Retrieval Solution. Target epitopes have variable tolerance to this exposure, with some signals remaining consistent throughout, whilst others are lost after a small number of iterations.

Each antibody was tested on a control section for signal attenuation by repeated stain and strip processes (number of repeats ≥ number of stains in mxIHC panel). Whole slide images (WSI) were captured after each staining cycle, then a conserved region that shows the target signal in the first stain is selected for all images and the stain is quantified as a relative area covered across multiple intensity bins. If the staining intensity was found to decrease after repeat *x*, the maximum position for the antibody in the Mx sequence was *x*−1.

**Multiplex image generation.** The WSI from each staining iteration was deconvoluted using the hue-saturation-density (HSD) model, implemented in Developer XD 2.7 (Definiens, Munich)[77], to create a 'pseudo-immunofluorescence' (pIF) image showing the blue (nuclear), brown

**Table 1 | Details of antibodies used for multiplexed Immunohistochemistry**

| Target | Clone | Supplier | Product code | Dilution | Ag retrieval pH |
|--------|-------|----------|--------------|----------|------------------|
| CD31 | JC70A | Agilent/Dako | IR61061-2 | 5x (RTU) | High |
| Pan-CK | AE1/AE3 | Agilent/Dako | IR05361-2 | 5x (RTU) | High |
| MCAM | Polyclonal | SIGMA/Merck | HPA008848 | 500x | Low |
| ACTA2 (αSMA) | 1A4 | Agilent/Dako | IR61161-2 | 1x (RTU) | High |
| POSTN | Polyclonal | SIGMA/Merck | HPA012306 | 50x | Low |
| CD34 | QBEnd 10 | Agilent/Dako | M716501-2 | 50x | Low |
| AOC3 | #393112 | R&D Systems | MAB3957 | 500x | Low |
| HSPA1A | 3A3 | Santa Cruz | sc-32239 | 250x | Low |

RTU product purchased at supplier designated "ready to use" concentration.

(registration) and red (transient marker) signals. This produced three raster images of floating-point values ranging from 0 to ~3. Storage and processing economy was maximised by converting float values to integers in the range of 8-bit images (1–256); for each pixel, the float value was divided by 1.5, then multiplied by 256 and rounded up to a maximum of 256. Each layer was then processed to identify topological maxima above the level of background signal, measured as the moving average over a $151 \times 151$-pixel window (Original-Background $<5 = 0$). For the red signal, an additional condition was applied to account for the similarity in the colour profile of red and brown (red signal < (0.43*brown signal) = 0). These pIF images, saved at a resolution equivalent to x5 magnification in tiff format, were registered using the *MultiStackRegistration* plugin[78] in Fiji[79]. The primary Mx image consists of a nuclear stain from iteration (i)1, registration stain from i1, target marker from i2 to i*n* and a nuclear marker from i*n*. For quality control (QC), an additional Mx image was generated consisting of nuclear marker staining from i1 to i*n*.

**Multiplex image analysis.** QC analysis was performed on the nuclear stain, to identify tissue loss or registration errors. A nuclear signal was identified from i1, then any nucleus with a signal missing in i2 to i*n were* marked as lost.

Mx analysis proceeded from the import of the result from QC analysis where the remaining nuclear signal was segmented to generate individual nuclear objects. These were then grown by $5.5\,\mu m$ radius to represent simulated cell regions. For each marker, a threshold of 7 was used to identify significant staining (this threshold is to exclude blush/bleed/non-specific staining) and the marker coverage for each cell was measured. This, plus XY coordinates for each cell were exported for histo-cytometry analysis.

**Stromal cell identification and histo-cytometry.** Histo-cytometry estimates of cellular expression levels were calculated as the fraction of pixels within simulated cell regions positive for staining once background levels had been subtracted. Stromal cells were identified by the absence of staining for Pan-CK, CD31 or MCAM and positivity for one of the fibroblast markers (ACTA2 [αSMA], POSTN, AOC3 and CD34). Then subpopulations were identified by which of the four markers was most highly expressed.

**Spatial analysis.** Tissue regions (Tumour, control and solid morphology regions) were annotated by a consultant pathologist (GJT and ECS) and point pattern data from the histo-cytometry analysis was used to assess region enrichment.

**Bulk transcriptomic data analysis**
**Whole-tissue transcriptome dataset download.** RNA-sequencing counts data from TCGA NSCLC cohorts was downloaded using the TCGAbiolinks (v2.16.4) R package[80]. Clinical data for these patients was downloaded using the TCGA clinical data resource[81]. Mutational status

and additional metadata were downloaded from cBioPortal[82]. Microarray and associated clinical data were downloaded from the NCBI GEO database (GSE72094, GSE31210, GSE68465, GSE4573, GSE157009, GSE157010) using the GEOquery (v2.56.0) R package[83]. For each LUAD dataset the molecular subtypes were calculated using the previously described nearest centroid approach[23].

**Digital cytometry using CIBERSORTx.** Digital cytometry was carried out using the online tool developed by ref. [27]. We used SMART-Seq2 single-cell RNA-sequencing data from the Travaglini et al. and Maynard et al. studies[18,34] to generate a single-cell reference sample matrix, consisting of cells from control and tumour tissues. This consisted of fibroblast subpopulations; pericytes and SMCs, identified during in silico fibroblast sorting (described above); endothelial, epithelial and immune cells, identified from the general annotation provided in the dataset metadata. To prevent memory errors, we limited the size of this reference matrix by randomly downsampling each cell-type to a maximum of 500 cells. This reference sample matrix was then used to generate a signature matrix file, using the default CIBERSORTx settings (Fig. S5a). Cell fraction imputation was then performed in absolute mode, using this signature matrix and 'S-mode' batch correction. RNA-seq mixture datasets (TCGA cohorts) were converted to counts per million and analysed without quantile normalisation. Microarray mixture datasets were analysed with quantile normalisation performed prior to deconvolution.

To test the digital cytometry results' accuracy, we constructed a pseudobulk dataset from the Maynard et al. scRNA-seq data, excluding cells used to generate the signature matrix[34]. Fibroblasts are significantly under-represented in scRNA-seq datasets[12]. Therefore, reconstructing bulk datasets from sample-specific cells meant that fibroblasts represented a very small fraction of many pseudobulk samples. To overcome this issue, we instead created a simulated pseudobulk dataset of 200 samples. For each sample, 1000 cells were randomly selected from three patient samples, then additional fibroblasts were added so that each sample consisted of between 1–50% fibroblasts. The pseudobulk dataset was then converted to counts per million and analysed as described above. Linear regression analysis was used to determine the accuracy of CIBERSORTx absolute scores as measures of fibroblast subpopulation abundance (Fig. S5b).

**Survival analysis**
Survival analysis was performed for each cohort, using the *survival* package in R. CIBERSORTx absolute scores were used to calculate the percentage of each fibroblast subpopulation present in each sample. These values were then used as continuous independent variables in Cox proportional hazards (CoxPH) regression modelling of overall survival rates. For categorical analyses, TCGA-LUAD cohort was used as the test dataset, and optimal dichotomisation thresholds were determined using the survival (v3.2-11) survminer (v0.4.9) R packages. Statistical significance was then assessed using a log-rank test (for

univariate analyses) or by CoxPH regression modelling (for multivariate analyses).

## Statistical analysis

All statistical tests carried out were two-sided and are described in the relevant figure legends, including the sample size for each experimental group. Exact *p* values are shown in the figures unless otherwise stated in the figure legends. All Boxplots are displayed using the Tukey method (centre line, median; box limits, upper and lower quartiles; whiskers, last point within a 1.5x interquartile range).

## Reporting summary

Further information on research design is available in the Nature Portfolio Reporting Summary linked to this article.

## Data availability

The raw scRNA-sequencing data generated in this study are available using the NCBI Gene Expression Omnibus database: GSE153935. Additional datasets required to reproduce the analysis and figures have been published on Zenodo (https://doi.org/10.5281/zenodo.7400873). These include a Seurat object holding scRNA-sequencing data and the results of our data integration for fibroblasts isolated from multiple human lung cancer datasets (used from Fig. 2 onwards in our paper); a dataframe holding the histo-cytometry results from our multiplexed immunohistochemistry (mxIHC) analysis performed on whole human lung cancer tissue sections; and further '.Rdata' files required for readers to reproduce the paper's analysis and figures. Source data are also provided with this paper as indicated in the figure legends. The publicly available datasets used in this study are available from the following sources. Bulk tissue RNA-seq (TCGA-LUAD/LUSC[24,25]) from the NIH-NCI Genomics Data commons (https://gdc.cancer.gov/access-data). The publicly available bulk tissue microarray datasets are available from the NCBI Gene expression omnibus (GEO; https://www.ncbi.nlm.nih.gov/geo/) under the following accession codes GSE72094[51], GSE31210[52], GSE68465[53], GSE4573[54], GSE157009[55] and GSE157010[55]. The publicly available NSCLC scRNA-seq datasets are available from various online repositories: the Kim et al. study data were available from NCBI GEO under the accession code GSE131907;[16] the Qian et al. dataset is available on the ArrayExpress database at EMBL-EBI under accession codes E-MTAB-6149 and E-MTAB-6653;[32] the Travaglini et al. dataset is available on the synapse database under the accession code syn21041850;[18] the Maynard et al. dataset is available as an NCBI BioProject under the accession code PRJNA591860 and processed data were accessed at the following link (https://github.com/czbiohub/scell_lung_adenocarcinoma)[34], the Bischoff et al. dataset is available as a Code Ocean capsule under the following https://doi.org/10.24433/CO.0121060.v1[31]. The publicly available IPF dataset used is available from NCBI GEO under the accession code GSE136831[15]. The publicly available PDAC dataset is available as an NCBI BioProject under the accession code PRJCA001063[49]; the publicly available CRC dataset is available from NCBI GEO under the accession code GSE132465[50]; the publicly available HNSCC dataset is available from NCBI GEO under the accession code GSE103322[29]. Source data are provided with this paper.

## Code availability

All coding was performed using the publicly available packages cited, in R (v4.0.2). For further details, the R scripts used are available on Github (https://github.com/cjh-lab/NCOMMS_NSCLC_scFibs.git).

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

## Author contributions

C.J.H. and G.J.T. contributed to the study conception, data interpretation and drafted the manuscript; C.J.H., C.H.O., M.J.J.R.-Z. and G.J.T. contributed to the study design; C.J.H., S.W., M.J.E., M.A.L., W.Y.P., J.T., R.P., L.M.K., S.J.C., E.C.S., J.W., A.A., E.W., C.H.O. and M.J.J.R.-Z contributed to data acquisition; C.J.H., S.W., M.E. and M.J.J.R.-Z. contributed to data analysis.

## Competing interests

The authors declare no competing interests.
