## [Peer Review File · Nature Communications]

Single-cell analysis reveals prognostic fibroblast subpopulations linked to molecular and immunological subtypes of lung cancerREVIEWER COMMENTS

Reviewer #1, expert in lung cancer genomics and molecular subtypes (Remarks to the Author):

This study reports the identification of 4 new transcriptomic subtypes of fibroblasts in normal lung and lung cancers, using publicly available single cell RNAseq data of lung adenocarcinoma (LUAD) and squamous cell carcinoma (LUSC) and control samples, plus data generated in their own lab. The authors then investigated the relevance of these fibroblast subtypes in patients with pulmonary interstitial fibrosis (IPF), pancreatic ductal adenocarcinoma (PDAC) and Head and Neck squamous cell carcinoma (HNSCC). They then explored the relevance of these subtypes in TCGA lung cancer bulk RNAseq data to demonstrate association of these fibroblast subtypes with previously reported lung cancer molecular subtypes, histological subtypes, immune-associated subtypes. Finally they investigated the association of the myofibroblast and alveolar fibroblast subtypes with NSCLC patient prognosis, using 4 published RNAseq/microarray datasets. The authors concluded that fibroblast can be heterogeneous and organ-specific and the findings could improve patient stratification and refinement of strategies for therapeutic targeting of fibroblasts in lung cancer (line 644-645).

General comments: This study is largely composed of 2 parts. The first part is analysis of scRNAseq profiling of their own 5 LUAD, 7 LUSC and 6 control (corresponding normal lung) samples. This analysis demonstrated the identification of 5 transcriptomic clusters, which appear distinctly associated with LUAD, LUSC or control. The result of this part of the study is largely preliminary due to small sample size and without further validation in independent datasets. The second part of the study is on meta-analysis of data combining their own and others that are available publicly. This analysis resulted in identification of 4 transcriptomic fibroblast subtypes. This part of the study is mainly in silico bioinformatics exercise, with multiple interesting but not robustly validated and mostly association results between the fibroblast subtypes and previously reported histological (Fig. 5Sd) and molecular subtypes (Fig.5a) of LUAD/LUSC, putative immune subtypes (Fig.5b and 5e), TP53 genotype (Fig.5c), non-neoplastic IPF (Fig.4b-d) and other cancer types (PDAC & HNSCC). The most interesting and possibly clinically relevant finding is the association of myofibroblast and alveolar fibroblast signatures with significantly differing prognosis in LUAD. However, greater details are necessary in the description and reporting of the latter analysis to assure that the results are robust and have potential as clinically useful prognostic markers.

Major issues:

1. While data from Part 1 (Target Lung in Fig.3a) is included in Part 2 analysis, there is no attempt to validate the findings of part 1 in other published datasets. Thus, the robustness or novelty of Part 1 findings are unclear.

2. MxIHC validation is attempted in Part 1, but there is no attempt to do the same for the new fibroblast subtypes identified in Part 2.

3. Table S1 provides information (fold-change) on expression of genes that are associated with the 4 types of fibroblasts. However, there is no explanation on what the data represent, e.g. FC to what? Furthermore, the statistical methods or formula that are used to classify patient tumors according to fibroblast types in the 4 expression datasets (Fig. 6 and S6) are unclear. It is also unclear how the combination subtypes (e.g. Myo.High/Alv.Low, etc) are identified (Fig.6). These details are critical for other investigators to reproduce or validate the prognostic findings independently.

4. Details of multivariate analysis for result shown in Fig.S6e should be provided. It is unusual to show survival outcome results of multivariable analysis using Kaplan-Meier plots (Fig.S6e and h). What variables were included in the multivariate analysis?

5. The Prognostic association of myofibroblast signatures appears consistent across 4 datasets (Fig.S6), less so for alveolar type signature. However, it is unclear how these signatures add to or compare to other prognostic signatures previously reported in NSCLC.

6. Fig.5c and 5d show correlation association of various fibroblast subtypes with “gene modules”. While this might be interesting from bioinformatics point of view, the biological meaning of these associations is unclear. In fact, the lack of biological validation of genes (even selective ones) that are associated with the various fibroblast subtypes renders the results as mainly hypothesis generating. Some biological validation is especially important when new type of CAF (e.g. stress-response) is described.

7. There should be more discussion on how the findings described in this study compare to or contrast with results reported in other studies.

Additional issues:

1. The role of CAFs in cancer progression and as therapeutic target is highly controversial. Many papers including editorials have been published on the subject. Yet, the authors only cited 3 references in the opening paragraph, all being selective and do not reflect the controversies. This is a major deficiency.

2. Fig.1A: more information on the samples used for the scRNAseq analysis is necessary, including patient demographics and histological features/subtypes of the LUAD.

3. Fig. 1b and S1a: Is the tSNE plot shown that of one sample or data from all samples combined? Since Fig. S1b shows marked heterogeneities in the composition of various cell types across samples, of even among the same sample types (i.e., normal, LUAD, LUSC), how are the heterogeneities integrated or considered in results shown in Fig.2, S1 and S2?

4. Nine samples with high aSMA were used for validation by multiplex IHC, are these samples the same ones used in the scRNAseq analyses? If not, what are they and their pathological features? How many are LUAD and LUSC, respectively?

5. Fig. S2d, 2j and 2m should show the H&E images for better histological assessment of these sample. As shown, it is not possible to know what they represent.

6. Fig. 2k: unclear whether the fractions shown represent those observed in one samples or multiple samples of same tumour types?

7. Fig.2e and 2j. Clusters 1 and 4 express iCAF markers (IL6, CFD, CCL2). However, these clusters are found in control normal lung tissue. This suggests the control samples may not be truly normal lung tissue. Have the control tissues been validated as showing normal lung by histology? It is not uncommon that non-neoplastic lung parenchyma in lung cancer resection specimens may show various inflammatory changes.

8. Fig. 2h and 2i: SERPINE1 and CXCL12 expression is substantial in clusters other than cluster 2, making them suboptimal as defining markers for cluster 2 (line 132-133).

9. MxIHC shows that approx. 30% (LUSC) and 20% (LUAD) of fibroblasts are double positive for SERPINE1 (cluster 2) and POSTN (cluster 0). What do these double positive fibroblasts represent?

10. Fig.S2c: The GSE plots of SERPINE1+ fibroblasts show pathways for oxidative stress and hypoxia, yet Fig. 2i shows SERPINE1+ fibroblasts present equally in control, margin and tumour. What is the possible biological explanation for this observation?

11. Fig.2m: Without H&E histology image, it is not possible to know what type of tissue is dominated by SERPINE1+ cells, especially in the context of pATM co-localization.

12. Pg.12, para 1: Fibroblasts refer to mesenchymal cells that produce collagen, and telocytes have been distinguished from fibroblasts by its lack of collagen-synthesis function. CD34 is a classical marker of endothelial cells. Labeling CD34+ telocytes as a type of fibroblast will create confusion in scientific literature. Has there been prior justification for this?

13. Pg 12, para2: Pathologically, myofibroblasts are seen only in tissue undergoing damage-repair process, and not seen in normal lung tissue. Therefore, the term is inconsistent with being a component of normal lung tissue, unless the “control” tissue the authors refer to are not normal lung tissue.

14. Pg.13, para 2: The authors stated that their “stress-response fibroblasts” are characterized by genes predominantly expressed by alveolar macrophages. Macrophages are definitely NOT fibroblasts, thus labeling this population as “fibroblasts” is not justified.

15. Fig 5a: The dotted line represents the median across all groups. Does “all groups” also include the control/normal samples? As there are significant differences between normal and tumours, for comparison between tumour molecular subtypes, the statistical comparison between each subtype must be to the median of only tumour tissue, LUAD and LUSC separately.

16. As additional expression datasets are available for LUAD (4 cohorts: n=1646) and LUSC (2 cohorts, n=631), validation of correlation between fibroblast subpopulations and molecular subtypes should also be possible in these independent datasets.

17. Pg. 21: differences in abundance of fibroblast associated with TP53 genotypes (WT vs MT) need validation in independent tumour cohorts (Fig S5c). The same applies to histological subtypes (Fig. S5d)

18. Correlation between fibroblast subpopulations with “immune subtypes” in cancer also lacks robust validation.

19. Line 477-478: Unclear why myofibroblasts, which are non-epithelial thus should not express keratins, should be associated with basal epithelial cells. The entire paragraph (line 479-485) does not provide new insight into the biology of lung cancer, thus is meaningless.

20. Pg. 12: the description of CD34+ fibroblasts should be supplemented with a breakdown by lung cancer types, as is done for all other subgroups.

Reviewer #2, expert in multiplex immunohistochemistry (Remarks to the Author):

In the manuscript entitled “Single-cell analysis reveals prognostic fibroblast subpopulations linked to molecular and immunological subtypes of lung cancer,” Hanley, Waise and colleagues characterize the transcriptomes of fibroblasts from 12 samples of lung cancer (7 SCC, 5 adenocarcinoma) and identify four major subpopulations. The authors find associations between fibroblast subtypes and clinical/molecular parameters and with patient survival.

There appear to be some discrepancies between the results from the authors’ original scRNAseq dataset and those derived from the bulk RNA-seq deconvolution using Cibersort. These points require further attention (see section below on Cibersort). In addition, the four-plex tissue imaging was not optimally leveraged. The imaging was technically well-performed and sufficiently analysed. However, the analysis of only nine samples was essentially only a validation of the scRNAseq work and therefore the study failed to leverage the power of multiplexed imaging which is in principle best used to study many additional samples and therefore extend the findings well beyond the few samples used in the discovery cohort (i.e., not just for simple validation of the discovery results). It seems that with a marker set that would amply identify the four different fibroblast subpopulations discovered in the scRNAseq data, the authors would have been able to perform (or at least confirm some of) the analyses in Figure 4-6 using tissue cohorts and hence generate more confidence in the conclusions (particular because some of the results from the scRNAseq discovery cohort seem to conflict with the reported results from the bulk deconvolved analysis).

Multiplexed imaging:

Comparison of sc-RNAseq data with multiplexed imaging:

- On Line 143, the authors state that they identified subsets of stromal cells “consistent with our sc-RNA-seq analysis” ... it is not entirely clear what consistent means in this particular context. If this pertains to cell state calling for example it would help if the authors were more explicit if the fraction of each subset of cells was consistent on a per sample level between the scRNA-seq data and the cell state calls made in the imaging data (using in a simple scatter plot).

- From the mapping of transcript expression levels onto the UMAP representation, it appears that POSTN+ CAFs are mostly in cluster 0 which is predominantly composed of CAF from LUAD and that SERPINE1+ CAFs are mostly expressed in cluster 2 which is predominantly composed of CAF from LUSC (or at least that is what it seems until the cibersort analysis) ... but the multiplexed imaging analysis and quantification shown in Figure 2k does not show a particularly strong enrichment of POSTN+ CAF in LUAD vs LUSC and SERPINE1+ CAF in LUSC. Do the authors see this subtype specific enrichment in LUAD

vs LUSC using the data utilized in the subsequent meta-analysis ... i.e. thereby implicating the presence of iCAFs in LUSC?

- In addition, it is confusing that the SERPINE1+ aSMA+ double positive populations were present in equal proportions in both normal and tumor regions; it seemed that the hypothesis from the scRNAseq data was that those iCAFs were going to be largely restricted to tumor CAFs (in cluster 2, in LUSC)

- Please see Technical/Minor comments below for additional questions on multiplexed imaging.

Cibersort Analysis

- Digital cytometry is an obscure term – it was unclear in the abstract if that was referring to the multiplexed immunohistochemistry imaging and did not become clear until Fig. 5 that it applied to cibersort analysis.

- The cibersort analysis seems to yield contradictory information about the relative prevalence of each fibroblast population in specific tumor subtypes when compared to the original scRNAseq dataset. Example 1: Line 119: C1 cells were more common in LUSC and Line 115-116: C0 cells were predominantly isolated from LUAD ... compared with Line 387: myofibroblast abundance (i.e., C0 cells) was significantly increased in LUSC and Line 390: the majority of LUSC tumours were negative for CD34+ teleocytes (i.e., C1 cells). That seems to be the opposite result. This discrepancy is rather confusing and calls into question the subsequent analysis based on cibersort proposing links between fibroblast populations and molecular subtypes of LUAD, attempts to make links with oncogenic drivers, analysis of immune subtypes in the data from Thorsson et al., network analysis and outcome analysis. In this regard, it would seem that it would help if the authors had bulk-RNAseq data from the same samples that they also analyzed with scRNAseq ... to see how closely the cibersort results match with the single cell measurements. Without confidence in this, it is difficult to have confidence in many of the conclusions.

- It seems to me that the type of analyses that the authors perform using cibersort would be better pursued using a multiplexed marker panel in tissue cohorts of well-annotated samples; or that the authors need to reconcile the seemingly conflicting observations noted above. It seems that at least a portion of the results would need to be confirmed/validated in actual tissue. Below is presented a panel of cell state markers that might be useful for that type of tissue analysis.

- The enrichment analysis performed for morphologic subtypes is also confusing because it is well known that these different morphologies are highly intermixed and co-exist within individual tumors, so a proper assessment of enrichment of fibroblast subtypes within distinct lung cancer morphologies very

likely would need the type of spatial information that is derived from tissue imaging rather than inferred data from bulk RNA-seq applied to tumor types with unclear portions of very likely intermixed morphologies. The main conclusions from the RNA deconvolution work should be confirmed in at least a few tissue samples.

Meta-analysis:

- The manuscript was quite clear through figure 2 but became harder to follow starting with the meta-analysis section because of the choice for presentation. Starting at Line 221, the authors state that the four major populations identified in the meta-analysis were analogous to the fibroblast clusters identified in the original analysis ... The way it is presented seems more like a proclamation than a conclusion based on an inductive argument that a reader can readily follow. The reasoning is ultimately presented, but it is not easy for the reader to follow. A first thought is that the prior clustering identified five clusters (0-4) and now we are being told that there are only four fibroblast subtypes ... presumably C3 did not map neatly or was integrated into one of the subtypes? I think that to make this section more readily interpretable to the reader that the links should be made between the prior clusters and the new terms (perhaps with a visual aid) and then the differences in gene expression can be investigated and presented in detail as currently done.

- Another confusing point is that the meta-analysis indicates that alveolar fibroblasts (C4) are the largest fraction of fibroblasts and the CD34+ telocyte population (C1) is the second largest fraction, however, in the original scRNAseq analysis performed by the authors, C4 was the smallest fraction and C1 was the second smallest, and so on, with C0 and C2. One is left to wonder why the fractions are apparently inverted. Are the samples in the discovery set simply outliers?

- It seems premature to state that alveolar fibroblasts are “organ-specific” based on the analysis as presented from just a few different tumors (PDAC, HNSCC, and IPF). A statement like this would benefit from the development of a panel of antibodies that cover the four fibroblast subtypes (C0: POSTN, C1: CD34, C2: SERPINE1 or perhaps a more subtype specific markers; C4: perhaps ITGA8 and GPC3) that can then be applied to multiple tissue microarrays of normal tissue and of a wide array of tumor types.

Technical/minor details

- It would help if the authors provided more detail regarding the tissue imaging; it seems apparent that whole slides were analysed but this should be stated explicitly in the methods section.

- It would be helpful if the authors explained in the methods why they focused their analysis on samples with “high aSMA expression” ... presumably the other 3 tumor samples that were omitted also had

sufficient stroma to analyze; what was the cutoff for excluding a case? What was the breakdown of histologies (LUSC vs LUAD) for the nine cases that were included in the imaging experiment.

- Describe in more detail the main types of errors that were observed when applying the stromal classifier to the entire imaging dataset (e.g., were these errors predominantly cell fusions, cell fissions, entirely missed cells, misclassification as tumor, was there a bias toward errors in needle-shaped fibroblasts or in more oval-shaped fibroblasts, etc.)

- What is the source of control tissue for comparisons in Fig 2i; Fig S2f shows that all three components were present in one sample ... was that the case for all nine samples; were all samples included in the control, tumor border and tumor analysis (did all of the samples have normal neighboring tissue). Control tissue is probably not the best term; more appropriate to label it adjacent normal lung or adjacent non-neoplastic lung.

- When referring to the multiplexed images it seems that it would be more appropriate to refer to aSMA (the protein) rather than the gene name ACTA2.

- It would be helpful if the authors show low power images of the H&E from all nine samples in the supplemental figures

- In Figure 2m, it would help to see the phospho-ATM analysis being reported for each of the four fibroblast categories (shown in Fig 2k) separately. i.e. is pATM higher in SERPINE1+ fibroblasts regardless of whether they express POSTN?

- iCAF cells are often reported to express PDGFRa ... is that also the case in lung?

- The challenge with the iCAF finding in normal lung is that 'control' tissue was from surgically resected lung from cancer patients (not directly cancer associated).

General comment:

It is hard to reconcile the fibroblast subtypes presented in this manuscript data with several other subtypes of fibroblasts that have been proposed in other organs systems such as mesenchymal stem cell derived fibroblasts mscCAF and antigen presenting fibroblasts apCAF (e.g., PMID: 32393771).

Reviewer #3, expert in fibroblast characterisation and single cell sequencing (Remarks to the Author):

Thank you for the opportunity to review this manuscript. This study represents an important addition to the growing field about fibroblast heterogeneity. The authors have studied difficult-to-obtain human material, have substantiated their findings with informative tissue immunostaining giving spatial perspective, and they have made the rare comparison between CAFs and other disease-associated fibroblasts. The final figure shows exciting data that the abundance of specific subpopulations is associated (i.e. could predict) outcomes.

In the abstract, ideally the survival implications of the different fibroblast populations would be specified. Currently it is vague.

The introduction provides a good background of the current understanding and the gaps in knowledge. In the final line (100), adjust “identifying” to “inferring subtype specific functions”.

Figure 1b – legend, indicate that this is pooled data representing all tissue types studied (or specify)

Figure 1c legend – define AT1/AT2

In the figure 1/S1 results text (or discussion), comment on the highly variable proportions of different cell types (particularly fibroblasts) across the samples.

S1A – the red/orange colours are too similar for the red/green colour blind (plasma/proliferating/B cells). This is ok in B when they are stacked in order, but very difficult to see on the tSNE. Are the proliferating cells representing a range of lineages? Are there other markers (or lack of epithelial/tumour markers) to provide insight?

S1D – adjust legend placement closer to plot

Figure 2 – The ACTA2 expression within the 0 and 2 CAF clusters is variable, with a notably bottom-heavy violin plot. With this level of low/negative cells, it seems important to indicate that this is not a distinguishing feature. And for 2k, does this result change if you include all cells in clusters 0/2, irrespective of ACTA2 expression? What was the threshold to be deemed positive for ACTA? Is cluster 3 included here?

2j/m – please specify on the legend that this image represents a stroma-rich tumour (if that is the case)

2l – explain in the results or discussion the implications of the non-distinguishing presence of SERPINE1 (normal and tumour expression alike) to its use as a CAF stratifier. Although 2f shows the UMAP and 2l shows the comparison across sites by IHC, is there space to also show its expression on a violin plot?

Line 226 – to “estimate” functional properties

Line 282 – specify in which samples (normal, tumour, or both)

Line 291-292 – check grammar

Figure 3 – is it possible to predict a “shift” from the telocyte population in health to the expansion of the stress-response population in tumours (are they pre-cursors, for example)?

Check caps in F

S3d/e and S4 – “myoCAF” signature, which is what was used earlier in text (instead of myCAF)

S3c – interesting lack of functional enrichment in LUSC. In health there seems to be clear functions, in LUAD the two CAF populations have real ECM features, but in LUSC, undistinguished.

Lines 335-340 – do these cells have an inflammatory signature across sites as was observed in the lung in this study? Because of the strong interest in inflammatory fibroblasts, their ubiquity, and abundance in healthy tissue, and a pretty consistent marker (CD34) is anticipated to be of interest and worth emphasizing.

Line 343 – check semicolon – emphasize the interpretation here (e.g. indicating something special about cancer)

Figure 4i, is there not a control? Seeing the shift in cell counts from healthy to diseased is very informative and notably absent.

Figure S4b legend, re-specify this is from the PDAC data (assuming it is)

Figure 5 - Associated text tells a relevant and interesting narrative but the legend needs improvement (and/or asterisk placement/box plot organisation) to understand the comparisons and where the important enrichments lie (maybe use of letters/symbols is needed?). It is felt that the reader needs to refer back to the text in the main body of the manuscript to appreciate this. B is somehow easier to follow in the absence of text explanation.

437 – specify “dampen”?

483 – citation needed for comparison between alveolar and papillary.

Figure 6 – very nice data (could potentially be more space-efficient if needed). Is there a threshold of abundance needed for this prediction (thinking back to the high variability in proportions in Figure 1)?

583 – or maybe this is due to their relative lack of commitment to a function (as indicated by the spider plots)

624 – Consider: can “have” an inflammatory phenotype (rather than develop)

Paragraph starting 632 – as mentioned above, please mention the implications of SERPINE1 also being expressed in non-tumour fibroblasts.

Reviewer #4, fibroblast characterisation and single cell sequencing (Remarks to the Author):

Reviewer comments to Hanley et al. Single-cell analysis reveals prognostic fibroblast subpopulations linked to molecular and immunological subtypes of lung cancer.

Summary:

In their manuscript, Hanley and colleagues analyzed cancer associated fibroblasts (CAF) from non-small cell lung cancer patients using single-cell RNA sequencing (scRNA-seq). From the authors' own scRNA-seq experiments, they define four distinct fibroblast populations, including two specific CAF populations. They confirm previously described lung fibroblast as well as CAF populations and define a stress-related CAF population. They validate their finding of CAF subpopulations using multiplexed immune-staining, pathologist scoring and machine learning for histological analysis. The authors integrate their scRNA-seq data with published lung datasets for extended validation and analyze the organotypicity of the identified fibroblast subtypes.

Further, the authors use a digital deconvolution tool (digital cytometry) on bulk RNA sequencing datasets from lung cancer patients to reveal the cellular composition of the respective samples. The authors correlate the fibroblast composition of the tumor samples with pathological / histological tumor classifications and clinical parameters to reveal specific overrepresentation of fibroblast subtypes in certain cancer types. From the fibroblast abundance-analysis the authors can stratify lung adenocarcinoma patients into prognostic groups that correlate with overall survival. Taken together, the authors illustrate the power of scRNA-seq for deconvolution of bulk transcription datasets, and highlight the importance of decoding fibroblast heterogeneity within lung cancer for beneficial patient stratification and possible therapeutic treatment.

General comments:

Overall, the study is well designed and data analysis is solid, applying state-of-the-art techniques. The authors use single-cell transcriptomic data for the deconvolution of clinical bulk RNA-seq datasets from lung cancer patients, and thereby provide important information about the cellular composition of the patient samples. However, considering the depth of single-cell data, it feels as if the analysis, as presented in the manuscript remains rather generalized. The authors use plenty of publically available databases/-sets which at least for this reviewer were difficult to follow and understand the rationale behind which dataset(s) were chosen for the respective analysis. While the results are presented coherently in each individual figure, it is less clear how the wealth of results are connected to each other.

Specific comments

Regarding Figure 1-2:

1. It is unclear why the authors choose to do the fibroblast subtype analysis on their own scRNA-seq dataset, and not directly using the integrated dataset.
2. Further, regarding the usage of SERPINE1 as a marker for a specific CAF subset the data are somewhat contradictory between the antibody staining and the scRNA-seq results. The authors propose SERPINE1 as a marker for their C2 subset mainly derived from LUSC samples (Fig 2a-b), however in their tissue analysis the SERPINE1+ population is found to almost equal amounts in the both LUAD and LUSC (Fig 2k-l). One reason for this can be, that cells in C2 are derived from almost exclusively one sample (Fig S2a), which decreases the validity of C2 cells as a representative subtype. Additionally, both markers, POSTN and SERPINE1 (protein: PAI-1), that are used to determine cell subset tissue distribution are secreted proteins, and thus less appropriate to identify the producer cell. The authors should consider to use marker proteins that are intracellularly or membrane bound. Alternatively, multiplexed in situ hybridization (e.g. RNAscope) could be employed (as validation).
3. In Fig S2f, the authors should explain how can it be that the population of SMA+SERPINE1+ cells is found in regions where no SMA+ cells are highlighted (density plot, compare first image and last image in second row)?
4. The seventh LUSC sample contributes only marginally to the tumor specific clusters, but instead to a comparable amount as the normal tissue samples to cluster C1 and C4. Should this sample still be considered as tumor sample?

Regarding Figure 3:

5. As mentioned above, it is surprising that the authors do not use the fibroblast subpopulation result from their meta-analysis for marker gene selection and subsequent tissue analysis. Further, it is unfortunate that the authors keep their single cell analysis restricted to the four major subpopulation clustering (alveolar, telocytes, myofibs and stress). For example, all four subpopulations contain cells collected from normal as well as tumor samples (Fig. 3c) and the intra-cluster differential expression between normal fibroblasts and tumor-derived fibroblast would be relevant analysis. In relation to these data, how do the authors reason about the very low amount of alveolar FB from LUSC samples, what is the underlying cause?
6. The authors should perform additional analysis, such as pseudo-time that have the potential reveal the relationships of the defined subpopulations, intermediate cell stages as well as relevant differential gene expression pattern along the trajectory.
7. Further analysis beyond Gene Ontology categories or matrisome gene expression levels within the major subpopulations should be considered by the authors to increase the impact of the manuscript. It becomes more and more appreciate that fibroblasts and CAF exhibit a high level of inter- as well as intra organ heterogeneity, and thus the somewhat general analysis considering only the major subpopulations misses out on potentially important and relevant details. For reference on fibroblast heterogeneity see: Tallquist M, doi.org/10.1146/annurev-physiol-021119-034527, LeBleu & Neilson,

FASEB J DOI: 10.1096/fj.201903188R, and Sahei et al. Nature Reviews <https://doi.org/10.1038/s41568-019-0238-1>.

8. It is unclear what purpose the comparison to the myCAF and iCAF signatures fulfills. Further, all genes included in myCAF and iCAF signature should be disclosed for proper interpretation of the analysis presented in Fig S3d-e.

Regarding Figure 4:

9. The authors employ machine learning algorithm to predict the four identified major fibroblast subpopulations in other datasets, which is an interesting aspect and could possibly developed into an applicable tool for scRNA-seq analysis. However, to test the functionality or their prediction module, the authors use only three different datasets (one from lung and two from pancreatic or head and neck tumors), and only analyze the preselected fibroblast populations of the test datasets. Thereby, important controls to interpret the results are not provided. It would be important to see the predictive score for unrelated cell types, such as epithelial cells or endothelial cells. Further, closer related connective tissue cell types, such as mural cells (pericytes, smooth muscle cells) or chondrocytes should be analyzed as control groups to validate the algorithm. Generally, the data presented in Figure 4 and S4 does somehow lack a proper connection to the other results and the authors could consider to omit the data in this manuscript.

10. The statement that telocytes (CD34+ fibroblasts) and myofibroblasts are conserved across tissues is likely an overstatement, especially considering the limited amounts of tissues tested and the lack of proper controls to validate the efficacy of the prediction algorithm.

11. In relation, Fig 4e depicts predicted myofibroblasts in many clouds (clusters) of the PDAC UMAP landscape, which can be a sign of an imprecise categorizations, since the cells distributed to the different clouds in the UMAP landscape should exhibit specific molecular fingerprints. The visualization of the clustering result from the different datasets (IPF, PDAC, HNSCC), overlaid with the prediction would be informative as to what extent the prediction recapitulates the dataset-specific cluster determination.

12. Additionally, it would helpful for the interpretation to be able to see the genes that are used by the algorithm for the subtype prediction.

Related to Figure 5:

13. The authors use scRNA-seq data for deconvolution of bulk RNA-seq patient data from the TCGA, using the online tool CIBERSORTx. This is an interesting analysis with substantial high impact. However, it is surprising that the authors choose to use two datasets from other publications to generate the signature matrix which serves as the bases for the deconvolution, instead of their (whole) meta-analysis

dataset; please explain. For proper interpretation of the deconvolution, it is necessary that the authors reveal the genes that are included in the signature matrix (Fig S5a).

14. After successful decoding of fibroblast subtype abundance in clinical tumor samples, the authors correlate the prevalence of each fibroblast subpopulations with a plethora of clinical parameters for both LUAD and LUSC samples. Unfortunately, it is difficult to comprehend the message from the results presented in Fig 5, other than that normal fibroblast populations (alveolar & telocyte) have a high abundance in control samples, and CAF subpopulations (myofibroblasts & stress) have a high abundance in tumor samples. It is unclear what relevance the statistical significance calculated in Fig 5a-b has, since (almost) all of the data groups are “significant”, please explain and revise. Please also indicate the number of samples (n) that are included in each block and used to calculate statistics.

15. Is this seemingly low difference of the two CAF subpopulations due to a too general classification (see comments above)?

16. Additionally, an analysis focusing on the difference of CAF subpopulations between the two different tumor types (LUAD & LUSC) has the potential to unveil important transcriptomic features within CAFs and raise the impact of the manuscript (see also comments below).

17. Similar to the Gene Ontology analysis, the network analysis presented in Fig 5c-d has a generalized character and the interpretation of the results is difficult. For example, what does the correlation of LUSC myofibroblasts with the “intermediate filaments and keratin” module mean, are the myofibroblasts suggested to express genes of the module, or does this allow conclusions about the overall tissue composition?

18. However, the analysis does reveal a difference in the “cell cycle” correlation between myofibroblasts from LUAD and LUSC, respectively. The authors do not follow up on this, but could probably investigate this in more detail.

Regarding Figure 6:

The clinical relevance lies outside of this reviewers’ competence. Thus, the comments should be regarded as from a non-expert on the clinical aspect.

19. The authors apply their fibroblast abundance scores to correlate with patient outcomes (overall survival). This analysis illustrates the power of scRNA-seq data and contains highly relevant and impactful results. It is interesting that only ratios between myofibroblasts and alveolar fibroblasts exhibit a prognostic relevance for LUAD patients. Obvious questions are, why the prognostic capacity is limited to LUAD and not present in LUSC as well, and why the stress-fibroblasts do not exhibit any prognostic capability. Can the authors comment, as well as investigate possible factors underlying these effects of LUAD myofibroblasts on overall survival?

20. The authors should control that these effects are independent of the overall tumor burden/stage (e.g. high burden = high number of myofibroblasts, low burden = high number of alveolar fibroblasts), or other parameters (multivariate analysis).

21. Further, since the authors have the single-cell transcriptional data available, an analysis of the LUAD myofibroblast population on the background of correlation to patient survival could reveal important gene expression patterns and would increase the impact of the manuscript. Would a more in-depth analysis (see comments above) of the myofibroblasts (and other) subpopulation lead to more defined prognostic model?

22. Since the stratification of patients into prognostic groups is suggested by the authors, from a clinically perspective, it is important to understand at what time the prognostic correlation can be performed, could the authors comment on this. Finally, what additional prognostic analysis could be introduced in accordance to the results from the myofibroblast-correlation?

Minor comments to the Figures:

23. labeling in Fig S2c should be changed to (C2)

24. labeling in Fig 3f is cut, please correct

25. in Fig 6e, f, g, add legends for bar plot segments

26. in Fig S6, why different order of spider plots between a & b?

27. generally, the authors could give more information in the Figure legends, especially in the legends of the supplementary figures, since they are not similarly limited in word counts

Response to Reviewers

Note to all reviewers:

We would like to thank all the reviewers for their insightful comments on our manuscript. We have performed substantial revisions to address the points raised, which have bolstered the evidence supporting our original manuscript's key findings. These changes include significantly improved validation across multiple techniques, demonstrating that our findings from scRNA-seq analysis are also confirmed by multiplexed immunohistochemistry (mxIHC) and CIBERSORTx mediated digital cytometry.

In the process of completing these revisions there is one noteworthy change to our interpretation of the data analysed, which is due to reviewer 1's astute request to improve the robustness of our data integration and clustering (additional issue 3). In addressing this comment, we have developed new approaches for quantitatively assessing batch effects (Figure S1a and S2a) and the biological relevance of different clustering solutions (Figure S2b). Leading to the implementation of both intra and inter-dataset batch correction prior to clustering. Implementing this more robust pipeline has shown that the "stress-response" population described in our original manuscript was not consistently identified across patient samples. Therefore, we no longer refer to these cells as a discrete subpopulation. However, it is still clear that the genes associated with a stress response are differentially expressed when fibroblasts are ordered in "pseudotime" through trajectory analysis (revised Figure 4), and therefore we now describe this phenomenon as a transient phase in fibroblast differentiation.

Reviewer #1, expert in lung cancer genomics and molecular subtypes (Remarks to the Author):

General comments: This study is largely composed of 2 parts. The first part is analysis of scRNAseq profiling of their own 5 LUAD, 7 LUSC and 6 control (corresponding normal lung) samples. This analysis demonstrated the identification of 5 transcriptomic clusters, which appear distinctly associated with LUAD, LUSC or control. The result of this part of the study is largely preliminary due to small sample size and without further validation in independent datasets. The second part of the study is on meta-analysis of data combining their own and others that are available publicly. This analysis resulted in identification of 4 transcriptomic fibroblast subtypes. This part of the study is mainly in silico bioinformatics exercise, with multiple interesting but not robustly validated and mostly association results between the fibroblast subtypes and previously reported histological (Fig. 5Sd) and molecular subtypes (Fig.5a) of LUAD/LUSC, putative immune subtypes (Fig.5b and 5S5e), TP53 genotype (Fig.55c), non-neoplastic IPF (Fig.4b-d) and other cancer types (PDAC & HNSCC). The most interesting and possibly clinically relevant finding is the association of myofibroblast and alveolar fibroblast signatures with significantly differing prognosis in LUAD. However, greater details are necessary in the description and reporting of the latter analysis to assure that the results are robust and have potential as clinically useful prognostic markers.

We are pleased that the reviewer highlighted the clinical potential of the survival associations identified in our study; and that they found our manuscript to contain multiple interesting results. We would also like to thank the reviewer for identifying limitations to the statistical approaches applied in our original manuscript. In addressing these we have substantially improved the robustness of our findings. Specific points raised by this reviewer are addressed below.

Major issues:

1. While data from Part 1 (Target Lung in Fig.3a) is included in Part 2 analysis, there is no attempt to validate the findings of part 1 in other published datasets. Thus, the robustness or novelty of Part 1 findings are unclear.

We thank the reviewer for highlighting a need to improve the clarity and robustness of the analyses presented in the initial section of our manuscript. The aims of the first results section were twofold: 1) to describe the novel dataset generated in this study, which is provided as a resource to the research community; and 2) to demonstrate the need to identify (and exclude) mural cells in order to accurately characterise fibroblast heterogeneity. These aims are now both addressed in the revised Figure 1.

We feel the second point is particularly important as multiple recent studies using scRNA-seq to characterise fibroblasts from whole tissue homogenates have omitted this step, which has led to incorrect classification of mural cells as CAFs/myofibroblasts -due to shared expression of commonly used CAF markers (e.g. ACTA2). To improve the robustness of this analysis we used gene signatures recently identified by Muhl et al (PMID: 32769974) to discriminate fibroblast and mural cell clusters, in addition to the canonical markers used in our original analysis; this is now described in results page 7 para 2 and figure 1/S1). In our original submission the validation of this approach across other scRNA-seq datasets analysed was only described in the methods section and not shown in the results. To address this, we now include a representative example of the filtering and in silico isolation approach that we have implemented across all datasets analysed in this study (Figure S1f-g), demonstrating the robustness and necessity of this approach when analysing fibroblasts in complex scRNA-seq datasets generated from whole tissue homogenates.

The focus of the remainder of our manuscript is accurately characterising fibroblast heterogeneity. Therefore, to increase the power of our analyses, we combined the clustering of fibroblasts from our novel dataset with the integration and validation of these clusters in an additional six datasets (Figure 2), further described in the subsequent figures and results sections.

2. MxIHC validation is attempted in Part 1, but there is no attempt to do the same for the new fibroblast subtypes identified in Part 2.

We have significantly expanded our MxIHC validation of the fibroblast subpopulations. The design and implementation of this MxIHC panel is now described in results page 13 para 1 and in Figure 3. Data from MxIHC analysis are now presented throughout the manuscript to validate the transcriptomics analyses.

3. Table S1 provides information (fold-change) on expression of genes that are associated with the 4 types of fibroblasts. However, there is no explanation on what the data represent, e.g. FC to what? Furthermore, the statistical methods or formula that are used to classify patient tumors according to fibroblast types in the 4 expression datasets (Fig. 6 and S6) are unclear. It is also unclear how the combination subtypes (e.g. Myo.High/Alv.Low, etc) are identified (Fig.6). These details are critical for other investigators to reproduce or validate the prognostic findings independently.

We have increased the detail provided in the legends for all our supplementary tables and the details of fibroblast marker analysis are now provided in Table S2. We have also expanded the description of our approach to categorising patient samples for survival analysis to enable the

reproduction of these results. Details are provided in Figure 6, in the results sections (page 18 para 1-3) and in the methods section (page 29 para 1).

4. Details of multivariate analysis for result shown in Fig.S6e should be provided. It is unusual to show survival outcome results of multivariable analysis using Kaplan-Meier plots (Fig.S6e and h). What variables were included in the multivariate analysis?

We have revised this analysis to assess whether categorising patients based on fibroblast subpopulations is an independent prognostic variable after adjusting for age, gender and disease stage. These results are now presented in Figure 6i-j and S6c-d).

5. *The Prognostic association of myofibroblast signatures appears consistent across 4 datasets (Fig.S6), less so for alveolar type signature. However, it is unclear how these signatures add to or compare to other prognostic signatures previously reported in NSCLC.*

The aim of the study was to characterise fibroblast heterogeneity in NSCLC and examine whether particular phenotypes link to survival (suggesting a role in tumour progression) . We show that digital cytometry enumeration of particular fibroblast subpopulations links these cells to varied survival rates in adenocarcinoma and therefore provide insight into their role in disease progression. We also show that fibroblast phenotypes link to molecular and histological subtypes of LUAD, which are themselves prognostic. These findings are distinct from previous studies that have used a range of methods to identify prognostic signatures, typically involving statistical modelling to identify the optimal combination of genes to stratify patient survival rates, rather than analysing specific cell subpopulations within the tumour microenvironment. Therefore, we do not believe that the comparison between these approaches is appropriate within the scope of this study and could detract from the overall message of this manuscript.

6. Fig.5c and 5d show correlation association of various fibroblast subtypes with “gene modules”. While this might be interesting from bioinformatics point of view, the biological meaning of these associations is unclear. In fact, the lack of biological validation of genes (even selective ones) that are associated with the various fibroblast subtypes renders the results as mainly hypothesis generating. Some biological validation is especially important when new type of CAF (e.g. stress-response) is described.

We agree that these analyses are mainly hypothesis generating (a point also raised by reviewer 4) and have removed them from the manuscript.

7. There should be more discussion on how the findings described in this study compare to or contrast with results reported in other studies.

We have performed a comprehensive analysis of previously described fibroblast subpopulation gene signatures in the context of the subpopulations we have identified, which is now included in Figure 2i-j, Figure S2f and Table S4.

Additional issues:

1. The role of CAFs in cancer progression and as therapeutic target is highly controversial. Many papers including editorials have been published on the subject. Yet, the authors only cited 3 references in the opening paragraph, all being selective and do not reflect the controversies. This is a major deficiency.

We agree that there is significant literature discussing this topic. However, given the article guidelines it is not possible to adequately reference all the relevant studies. Therefore, we have re-written this section to emphasise that the utility of CAF targeting is still unproven, including references to a review that details ongoing and completed CAF targeting clinical trials and also research articles where preclinical models have called into question the viability of CAF targeting as a plausible approach to cancer therapy (page 5 para 1).

2. Fig.1A: more information on the samples used for the scRNAseq analysis is necessary, including patient demographics and histological features/subtypes of the LUAD.

These details have been added to Figure 1b and Table S1.

3. Fig. 1b and S1a: Is the tSNE plot shown that of one sample or data from all samples combined? Since Fig. S1b shows marked heterogeneities in the composition of various cell types across samples, of even among the same sample types (i.e., normal, LUAD, LUSC), how are the heterogeneities integrated or considered in results shown in Fig.2, S1 and S2?

In our original manuscript we adopted a common (but perhaps suboptimal) approach to determining whether batch effects were driving the clustering results by examining whether our clusters consisted of cells obtained from multiple samples. In response to the reviewer's comment, we have employed a more quantitative approach to examining potential batch effects. To do this we examined whether cells from potentially confounding covariates (e.g. the date samples were processed) had a significant increase in shared k-nearest neighbours, which is the metric used for clustering in Seurat, and designed an algorithm to test for this (described in methods section page 24 lines 632-637). This identified significant batch effects within both our dataset and the additional publicly available datasets analysed. Therefore, we implemented an intra-dataset integration using reciprocal PCA, prior to clustering and in silico fibroblast sorting, and then performed inter-dataset integration by canonical correlation analysis when preparing the integrated fibroblast dataset (described in methods section pages 24-25).

Notably this re-examination of the data showed that the stress response sub-subpopulation is no longer identified as a prominent and consistently identified cluster, but rather a transitional state activated as control tissue fibroblasts differentiate into myofibroblasts (trajectory analysis; Figure 4).

4. Nine samples with high α SMA were used for validation by multiplex IHC, are these samples the same ones used in the scRNAseq analyses? If not, what are they and their pathological features? How many are LUAD and LUSC, respectively?

5. Fig. S2d, 2j and 2m should show the H&E images for better histological assessment of these sample. As shown, it is not possible to know what they represent.

6. Fig. 2k: unclear whether the fractions shown represent those observed in one samples or multiple samples of same tumour types?

As described above (major point 2) the MxIHC data has been substantially revised and the changes requested in points 4-6 have been incorporated throughout the revised Figures and the clinicopathological details of the cohort analysed is now described in Table S6.

7. Fig.2e and 2j. Clusters 1 and 4 express iCAF markers (IL6, CFD, CCL2). However, these clusters are found in control normal lung tissue. This suggests the control samples may not be truly normal lung

tissue. Have the control tissues been validated as showing normal lung by histology? It is not uncommon that non-neoplastic lung parenchyma in lung cancer resection specimens may show various inflammatory changes.

We agree that inflammatory changes, commonly found in non-neoplastic lung parenchyma, are a valid explanation for why previously described iCAF markers are found in the control tissues and have now added a comment to this effect in the discussion (page 21 para 3). Throughout the manuscript the term 'control tissues' is used when describing these samples (i.e. these samples are not involved by tumour but should not be considered 'normal'). Samples were assessed by a pathologist during post-surgical sampling to confirm that they were non-neoplastic, however this did not preclude the presence of inflammatory changes.

We also examine the expression of inflammatory response genes in more detail as part of our trajectory analysis (Figure 4). We show that expression of these genes is significantly greater in control samples compared to tumour samples. These findings are a key reason why we avoided using the term iCAF to describe these cells in the context of NSCLC and we believe that as the reviewer suggests, the inflammatory response observed in these cells is likely to be the result of challenges to homeostasis within the lung but independent of these cells interaction with malignant epithelial cells.

8. Fig. 2h and 2i: SERPINE1 and CXCL12 expression is substantial in clusters other than cluster 2, making them suboptimal as defining markers for cluster 2 (line 132-133).

As described above, the improved and expanded integration approach has shown this population is not consistently observed across patients. All markers identified for the revised fibroblast subpopulations have been confirmed at a sample level and are detailed in Table S2.

9. MxIHC shows that approx. 30% (LUSC) and 20% (LUAD) of fibroblasts are double positive for SERPINE1 (cluster 2) and POSTN (cluster 0). What do these double positive fibroblasts represent?

10. Fig.S2c: The GSE plots of SERPINE1+ fibroblasts show pathways for oxidative stress and hypoxia, yet Fig. 2i shows SERPINE1+ fibroblasts present equally in control, margin and tumour. What is the possible biological explanation for this observation?

11. Fig.2m: Without H&E histology image, it is not possible to know what type of tissue is dominated by SERPINE1+ cells, especially in the context of pATM co-localization.

Each of these points are addressed or no longer applicable in the revised manuscript now that we have revised the MxIHC analysis of fibroblast subpopulations.

12. Pg.12, para 1: Fibroblasts refer to mesenchymal cells that produce collagen, and telocytes have been distinguished from fibroblasts by its lack of collagen-synthesis function. CD34 is a classical marker of endothelial cells. Labeling CD34+ telocytes as a type of fibroblast will create confusion in scientific literature. Has there been prior justification for this?

We agree that our original fibroblast subpopulation naming strategy could confuse the literature. In our revised manuscript we use terminology consistent with previous scRNASeq studies examining fibroblasts in normal and fibrotic lung tissue (PMIDs: 32317643, 33208946 and 33981032).

13. Pg 12, para2: Pathologically, myofibroblasts are seen only in tissue undergoing damage-repair

process, and not seen in normal lung tissue. Therefore, the term is inconsistent with being a component of normal lung tissue, unless the “control” tissue the authors refer to are not normal lung tissue.

Myofibroblasts are increased in tissue undergoing damage repair, but it is not true to state that myofibroblasts are not present in normal lung tissue. Multiple high-resolution analyses of cellular phenotypes in normal lung tissue have documented the presence of myofibroblasts in adult human and murine alveolar tissues (PMIDs: 32317643, 33208946). Similarly, myofibroblasts have also been found in the stroma surrounding normal intestinal mucosa (PMID: 21252048). It is noteworthy that the phenotype of the myofibroblasts that we have identified in normal tissue are phenotypically distinct from those found in tumour tissues (Figure 2h-j and S2g-h), clarifying the functional consequence of these differences will be the basis of future studies.

14. Pg.13, para 2: The authors stated that their “stress-response fibroblasts” are characterized by genes predominantly expressed by alveolar macrophages. Macrophages are definitely NOT fibroblasts, thus labeling this population as “fibroblasts” is not justified.

We have removed this section - the stress response cluster represents a fibroblast transition state (see response to the editor’s comments above)

15. Fig 5a: The dotted line represents the median across all groups. Does “all groups” also include the control/normal samples? As there are significant differences between normal and tumours, for comparison between tumour molecular subtypes, the statistical comparison between each subtype must be to the median of only tumour tissue, LUAD and LUSC separately.

We apologise for this error. Our analysis did compare to the median of all (control and tumour) groups. We have corrected this and re-run the statistical tests as advised comparing between groups within the NSCLC subtype only (Figure 7 and Figure S7).

16. As additional expression datasets are available for LUAD (4 cohorts: n=1646) and LUSC (2 cohorts, n=631), validation of correlation between fibroblast subpopulations and molecular subtypes should also be possible in these independent datasets.

17. Pg. 21: differences in abundance of fibroblast associated with TP53 genotypes (WT vs MT) need validation in independent tumour cohorts (Fig S5c). The same applies to histological subtypes (Fig. S5d).

We have extended our analysis accordingly (fig. 7 and S7).

18. Correlation between fibroblast subpopulations with “immune subtypes” in cancer also lacks robust validation.

Due to difficulties in accurately re-creating these immune subtype classifications across additional datasets we replaced this analysis with the examination of correlations between fibroblast subpopulations and LM22 immune cell populations, confirming our results across multiple datasets (Figure 7g and S7e).

19. Line 477-478: Unclear why myofibroblasts, which are non-epithelial thus should not express keratins, should be associated with basal epithelial cells. The entire paragraph (line 479-485) does not provide new insight into the biology of lung cancer, thus is meaningless.

As described above (major point 6), the results from this analysis have been removed.

20. Pg. 12: the description of CD34+ fibroblasts should be supplemented with a breakdown by lung cancer types, as is done for all other subgroups.

In the revised manuscript all subpopulations are described comparably in this section and now present the percentage of each subpopulation across subgroups as measured by multiple approaches in Figure 3e-g.

Reviewer #2, expert in multiplex immunohistochemistry (Remarks to the Author):

In the manuscript entitled “Single-cell analysis reveals prognostic fibroblast subpopulations linked to molecular and immunological subtypes of lung cancer,” Hanley, Waise and colleagues characterize the transcriptomes of fibroblasts from 12 samples of lung cancer (7 SCC, 5 adenocarcinoma) and identify four major subpopulations. The authors find associations between fibroblast subtypes and clinical/molecular parameters and with patient survival.

There appear to be some discrepancies between the results from the authors’ original scRNAseq dataset and those derived from the bulk RNA-seq deconvolution using Cibersort. These points require further attention (see section below on Cibersort). In addition, the four-plex tissue imaging was not optimally leveraged. The imaging was technically well-performed and sufficiently analysed. However, the analysis of only nine samples was essentially only a validation of the scRNAseq work and therefore the study failed to leverage the power of multiplexed imaging which is in principle best used to study many additional samples and therefore extend the findings well beyond the few samples used in the discovery cohort (i.e., not just for simple validation of the discovery results). It seems that with a marker set that would amply identify the four different fibroblast subpopulations discovered in the scRNAseq data, the authors would have been able to perform (or at least confirm some of) the analyses in Figure 4-6 using tissue cohorts and hence generate more confidence in the conclusions (particular because some of the results from the scRNAseq discovery cohort seem to conflict with the reported results from the bulk deconvolved analysis).

We thank the reviewer for their expert insight into improving the use of multiplexed immunostaining to enrich our manuscript. We have performed extensive revisions to the MxIHC analysis, providing additional biological insight and more robust validation of the transcriptomic findings. These results are initially described on page 13 para 1 and shown in Figure 3/S3, in addition these data are also used in figures 4, 5 & 7 to validate key observations from our scRNA-seq analysis.

Multiplexed imaging:

Comparison of sc-RNAseq data with multiplexed imaging:- On Line 143, the authors state that they identified subsets of stromal cells “consistent with our sc-RNA-seq analysis” ... it is not entirely clear what consistent means in this particular context. If this pertains to cell state calling for example it would help if the authors were more explicit if the fraction of each subset of cells was consistent on a per sample level between the scRNA-seq data and the cell state calls made in the imaging data (using in a simple scatter plot).

Using a revised mxIHC panel we performed the requested analysis, which showed a significant correlation between the fibroblast subpopulation abundance detected by scRNA-seq and mxIHC ($r=0.55$, $p=0.001$). However, this analysis does require the caveat that mxIHC and scRNA-seq cannot be performed on the exact same piece of tissue (with MxIHC subsequently performed on FFPE archived tissue from the same tumour). Therefore, given likely tumour heterogeneity, it should not be expected that the results of this analysis would necessarily be fully concordant. To avoid confusion, we have omitted the statement quoted in the reviewer’s comment and in the revised manuscript provided clearly comparable figures to demonstrate where scRNA-seq results have been validated through mxIHC analysis (e.g. Figures 3e/f and 7c/d)

- From the mapping of transcript expression levels onto the UMAP representation, it appears that POSTN+ CAFs are mostly in cluster 0 which is predominantly composed of CAF from LUAD and that SERPINE1+ CAFs are mostly expressed in cluster 2 which is predominantly composed of CAF from LUSC (or at least that is what it seems until the ciphersort analysis) ... but the multiplexed imaging analysis and quantification shown in Figure 2k does not show a particularly strong enrichment of POSTN+ CAF in LUAD vs LUSC and SERPINE1+ CAF in LUSC. Do the authors see this subtype specific enrichment in LUAD vs LUSC using the data utilized in the subsequent meta-analysis ... i.e. thereby implicating the presence of iCAFs in LUSC?

- In addition, it is confusing that the SERPINE1+ aSMA+ double positive populations were present in equal proportions in both normal and tumor regions; it seemed that the hypothesis from the scRNAseq data was that those iCAFs were going to be largely restricted to tumor CAFs (in cluster 2, in LUSC)

- Please see Technical/Minor comments below for additional questions on multiplexed imaging.

As described above (editor's comment and reviewer 1 additional point 3), we revised our scRNA-seq data processing approach introducing reciprocal PCA for intra dataset batch correction. This showed that the SERPINE1+ CAF population represented a transitional state and was not consistently identified as a separate subpopulation across multiple samples once the rPCA method of intra-dataset integration was implemented.

Cibersort Analysis

- Digital cytometry is an obscure term – it was unclear in the abstract if that was referring to the multiplexed immunohistochemistry imaging and did not become clear until Fig. 5 that it applied to ciphersort analysis.

We chose to use this term as this is what the CIBERSORTx authors used to refer to the deconvolution tool in their original paper. To clarify this point we have changed the use of “digital cytometry” to “digital cytometry implemented using CIBERSORTx” or equivalents throughout the manuscript.

- The ciphersort analysis seems to yield contradictory information about the relative prevalence of each fibroblast population in specific tumor subtypes when compared to the original scRNAseq dataset. Example 1: Line 119: C1 cells were more common in LUSC and Line 115-116: C0 cells were predominantly isolated from LUAD ... compared with Line 387: myofibroblast abundance (i.e., C0 cells) was significantly increased in LUSC and Line 390: the majority of LUSC tumours were negative for CD34+ teleocytes (i.e., C1 cells). That seems to be the opposite result. This discrepancy is rather confusing and calls into question the subsequent analysis based on ciphersort proposing links between fibroblast populations and molecular subtypes of LUAD, attempts to make links with oncogenic drivers, analysis of immune subtypes in the data from Thorsson et al., network analysis and outcome analysis. In this regard, it would seem that it would help if the authors had bulk-RNAseq data from the same samples that they also analyzed with scRNAseq ... to see how closely the ciphersort results match with the single cell measurements. Without confidence in this, it is difficult to have confidence in many of the conclusions.

To clarify, we did not intend to claim there were statistically significant differences between NSCLC subtypes in the novel scRNA-seq data (TLDS) generated in this study. This is because the TLDS does not have adequate power to detect such changes. To clarify this point, in the revised manuscript

we have changed the structure of the results such that associations between fibroblast subpopulation abundance and sample type are only described using the integrated dataset, which has a much larger sample size and therefore adequate power to assess these differences (Revised Figure 3e-g described on page 13 para 5). These results are consistent with what is observed using the CIBERSORTx digital cytometry method and we have added further validation of these findings by mxIHC as suggested by the reviewer.

- It seems to me that the type of analyses that the authors perform using cibersort would be better pursued using a multiplexed marker panel in tissue cohorts of well-annotated samples; or that the authors need to reconcile the seemingly conflicting observations noted above. It seems that at least a portion of the results would need to be confirmed/validated in actual tissue. Below is presented a panel of cell state markers that might be useful for that type of tissue analysis.

We performed rigorous testing of the CIBERSORTx deconvolution algorithm using bulk transcriptomic datasets simulated from single-cell transcriptomes, confirming that the abundance of each subpopulation is accurately imputed using this method (described on page 13 para 4 and Fig. S3e). Furthermore (as described in response to the point above), in the revised manuscript we also demonstrated that the CIBERSORTx results are consistent with the scRNA-seq analysis and mxIHC analyses (Figures 3e-g and 7b-d).

We chose to use CIBERSORTx applied to publicly available datasets in order to assess survival and other clinical correlations for multiple reasons:

- *This approach has enabled the analysis of over 2500 NSCLC samples from different continents.*
- *These samples were linked to not only clinical information, but also transcriptomic and genomic data allowing the enhanced insight gained from associations with molecular subtypes and TP53 mutations.*
- *We have been able to validate all our findings across multiple independent cohorts, processed by separate research institutes demonstrating that our findings are extremely robust.*
- *The use of these publicly available datasets also enables future studies to directly compare or build upon the findings presented in this study, which we have facilitated by collating all the relevant data from these analyses into Table S9*

- The enrichment analysis performed for morphologic subtypes is also confusing because it is well known that these different morphologies are highly intermixed and co-exist within individual tumors, so a proper assessment of enrichment of fibroblast subtypes within distinct lung cancer morphologies very likely would need the type of spatial information that is derived from tissue imaging rather than inferred data from bulk RNA-seq applied to tumor types with unclear portions of very likely intermixed morphologies. The main conclusions from the RNA deconvolution work should be confirmed in at least a few tissue samples.

We have significantly expanded our analysis of the relationship between fibroblast subpopulations and morphological phenotypes of LUAD. In order to enable consistent examination of associations between fibroblast subpopulations and differentiation (across the scRNA-seq, CIBERSORTx and mxIHC datasets), we used the grade assessment provided in each dataset's meta data; or by the predominant morphological subtype.

In addition, to address the point about the intermixed nature of LUAD tumours in regard to these morphological subtypes we also specifically examined the correlations between pathologist assessed

proportions of different morphologies and mxIHC quantification of fibroblast subpopulations (Fig.S7b); and confirmed that myofibroblasts are the predominant fibroblast subpopulation found in specific regions of the tumour classified as solid morphology (as shown in Supplementary Data 6-9).

Meta-analysis:

- The manuscript was quite clear through figure 2 but became harder to follow starting with the meta-analysis section because of the choice for presentation. Starting at Line 221, the authors state that the four major populations identified in the meta-analysis were analogous to the fibroblast clusters identified in the original analysis ... The way it is presented seems more like a proclamation than a conclusion based on an inductive argument that a reader can readily follow. The reasoning is ultimately presented, but it is not easy for the reader to follow. A first thought is that the prior clustering identified five clusters (0-4) and now we are being told that there are only four fibroblast subtypes ... presumably C3 did not map neatly or was integrated into one of the subtypes? I think that to make this section more readily interpretable to the reader that the links should be made between the prior clusters and the new terms (perhaps with a visual aid) and then the differences in gene expression can be investigated and presented in detail as currently done.

To improve clarity on the nomenclature used for fibroblast subpopulations, we have changed this section to describe overlap with previous studies (which were not yet peer reviewed at the time of our initial submission; described on page 9 para 2 and Figure S2c) and clarified terminology.

As described above, (reviewer 1, major issue 1), we have also changed the structure of the results section to clarify these points, with the fibroblast heterogeneity analysis limited to the integrated dataset; this provides sufficient power to enable population level analysis of gene expression. The original discrepancy between the number of clusters was due to the exclusion of mural cells when defining fibroblast subpopulations. To clarify this point we have expanded the associated results section, making the identification and exclusion of these cells a prominent feature of the data presented in figure 1 and added a section to the introduction (page 5 para 3) to explain the importance of this step.

- Another confusing point is that the meta-analysis indicates that alveolar fibroblasts (C4) are the largest fraction of fibroblasts and the CD34+ telocyte population (C1) is the second largest fraction, however, in the original scRNAseq analysis performed by the authors, C4 was the smallest fraction and C1 was the second smallest, and so on, with C0 and C2. One is left to wonder why the fractions are apparently inverted. Are the samples in the discovery set simply outliers?

This discrepancy was due to variation in the type of samples in the TLDS compared to the integrated dataset. In the integrated dataset a greater proportion of the fibroblasts were isolated from control samples, whereas the TLDS has a higher proportion of LUSC fibroblasts. As we show in Figure 3 d-f there are significant differences in the abundance of each fibroblast subpopulation between the tissue subtypes, which accounts for the apparent discrepancy presented in our original manuscript. This comment also highlights that assessing the abundance/proportion of a subpopulation across the whole dataset can be misleading as not all samples contribute the same number of fibroblasts. As such we now only present fibroblast subpopulation abundance as proportions of all fibroblasts isolated from a given sample to account for this variation.

- It seems premature to state that alveolar fibroblasts are “organ-specific” based on the analysis as presented from just a few different tumors (PDAC, HNSCC, and IPF). A statement like this would benefit from the development of a panel of antibodies that cover the four fibroblast subtypes (C0: POSTN, C1: CD34, C2: SERPINE1 or perhaps a more subtype specific markers; C4: perhaps ITGA8 and

GPC3) that can then be applied to multiple tissue microarrays of normal tissue and of a wide array of tumor types.

We agree that extra data would provide greater credence to the statement that these cells are “organ-specific”. To address this, we have expanded our scRNA-seq analysis to include colorectal cancer, which has provided similar results demonstrating that, compared to myofibroblast and adventitial subpopulations, alveolar fibroblasts show greater phenotypic divergence between the lung and other tissues. In addition, as the reviewer suggested we confirm these findings using mxIHC analysis of TMAs from multiple tissue types (Figure 5).

Technical/minor

details

- It would help if the authors provided more detail regarding the tissue imaging; it seems apparent that whole slides were analysed but this should be stated explicitly in the methods section.
- It would be helpful if the authors explained in the methods why they focused their analysis on samples with “high aSMA expression” ... presumably the other 3 tumor samples that were omitted also had sufficient stroma to analyze; what was the cutoff for excluding a case? What was the breakdown of histologies (LUSC vs LUAD) for the nine cases that were included in the imaging experiment.
- Describe in more detail the main types of errors that were observed when applying the stromal classifier to the entire imaging dataset (e.g., were these errors predominantly cell fusions, cell fissions, entirely missed cells, misclassification as tumor, was there a bias toward errors in needle-shaped fibroblasts or in more oval-shaped fibroblasts, etc.)
- What is the source of control tissue for comparisons in Fig 2i; Fig S2f shows that all three components were present in one sample ... was that the case for all nine samples; were all samples included in the control, tumor border and tumor analysis (did all of the samples have normal neighboring tissue). Control tissue is probably not the best term; more appropriate to label it adjacent normal lung or adjacent non-neoplastic lung.

These points are all addressed in the expanded MxIHC analysis (presented in Figure 3 and Table S6).

The revised and expanded MxIHC analysis included more exclusion markers (PanCK, CD31, MCAM). Therefore, we used a simpler algorithm for classifying cells utilising marker expression to distinguish fibroblasts as described in the updated methods section [starting on line 973]].

For comparison between control and tumour tissues we used pathologist annotated regions within whole-tissue sections, and similarly within tumour regions for identifying poorly differentiated regions.

- When referring to the multiplexed images it seems that it would be more appropriate to refer to aSMA (the protein) rather than the gene name ACTA2.

We agree that for some readers that are familiar with α SMA this may be more appropriate. However, this also has the possibility to confuse readers that are not familiar with α SMA as the product of the gene ACTA2. Therefore, we maintain that the use of ACTA2 in accordance with standard gene symbol formatting guidelines should serve to reduce ambiguity and have added α SMA in brackets when this is used to describe the protein.

- It would be helpful if the authors show low power images of the H&E from all nine samples in the supplemental figures

As requested, an additional Supplementary Figure has now been included with low resolution H&Es provided for each of the samples analysed by mxIHC (Supplementary Data 2).

- In Figure 2m, it would help to see the phospho-ATM analysis being reported for each of the four fibroblast categories (shown in Fig 2k) separately. i.e. is pATM higher in SERPINE1+ fibroblasts regardless of whether they express POSTN?

Due to the alterations described above, in response to reviewer 1 (minor issue 3), we have re-evaluated the contribution of stress response signalling to fibroblast heterogeneity identifying this as a transient phase in the transdifferentiation between control fibroblasts and tumour associated myofibroblasts. This analysis is now presented in Figure 4 and to ensure consistency between the scRNA-seq and mxIHC analyses we have opted for using HSPA1A staining to assess this in the different fibroblast populations. Results from which are now presented as suggested in a subpopulation specific manner (Figure 4i)

- iCAF cells are often reported to express PDGFRa ... is that also the case in lung?

PDGFRA is consistently expressed by all fibroblast subpopulations (shown in the violin plot below) and was not identified as a differentially expressed gene in any of our analyses

- The challenge with the iCAF finding in normal lung is that 'control' tissue was from surgically resected lung from cancer patients (not directly cancer associated).

We agree that use of the term 'iCAF' is problematic. However, the presence of iCAFs in non-tumour tissues is consistent with previous published studies describing iCAF phenotypes (PMID: 31197017). Notably, a recent analysis showed that inflammatory fibroblasts (including CAFs) may represent universal (adventitial) fibroblasts that have not undergone sufficient transcriptional change to constitute a new cell state (PMID: 33981032). We have added detailed analysis of differential expression of inflammation related genes in fibroblasts, showing that those expressing inflammatory genes are more common in control tissue (Figure 2j and 4g). We also elaborate on this point in the discussion providing reference to multiple studies that have described similar phenotypes in non-cancer pathologies (starting on line 764). We therefore avoid using the term iCAF to describe any of the subpopulations identified in this study.

General

comment:

It is hard to reconcile the fibroblast subtypes presented in this manuscript data with several other subtypes of fibroblasts that have been proposed in other organs systems such as mesenchymal stem cell derived fibroblasts mscCAF and antigen presenting fibroblasts apCAF (e.g., PMID: 32393771).

This is an important point and in the revised manuscript we have calculated module scores for many previously described fibroblast subpopulations (detailed in Figure S2f and Table S4).

Reviewer #3, expert in fibroblast characterisation and single cell sequencing (Remarks to the Author):

Thank you for the opportunity to review this manuscript. This study represents an important addition to the growing field about fibroblast heterogeneity. The authors have studied difficult-to-obtain human material, have substantiated their findings with informative tissue immunostaining giving spatial perspective, and they have made the rare comparison between CAFs and other disease-associated fibroblasts. The final figure shows exciting data that the abundance of specific subpopulations is associated (i.e. could predict) outcomes.

We would like to thank the reviewer for acknowledging the importance of our study to the growing field of fibroblast heterogeneity.

In the abstract, ideally the survival implications of the different fibroblast populations would be specified. Currently it is vague.

We have amended the abstract as requested.

The introduction provides a good background of the current understanding and the gaps in knowledge. In the final line (100), adjust “identifying” to “inferring subtype specific functions”.

We have modified the introduction and this sentence is no longer included.

Figure 1b – legend, indicate that this is pooled data representing all tissue types studied (or specify)
Figure 1c legend – define AT1/AT2

These legends have been updated accordingly.

In the figure 1/S1 results text (or discussion), comment on the highly variable proportions of different cell types (particularly fibroblasts) across the samples.

As suggested, we have added a comment to explicitly identify this variability within our dataset and the resulting impact on downstream analyses (page 9 para 1).

S1A – the red/orange colours are too similar for the red/green colour blind (plasma/proliferating/B cells). This is ok in B when they are stacked in order, but very difficult to see on the tSNE. Are the proliferating cells representing a range of lineages? Are there other markers (or lack of epithelial/tumour markers) to provide insight?

As we do not elaborate on these subpopulations further in this manuscript, to improve clarity we have limited our description of cellular subpopulations in the TLDS presented in Figure 1 and applied labels using a colour-blind friendly palette to the figures.

S1D – adjust legend placement closer to plot

This is no longer applicable in the revised figures.

Figure 2 – The ACTA2 expression within the 0 and 2 CAF clusters is variable, with a notably bottom-heavy violin plot. With this level of low/negative cells, it seems important to indicate that this is not a distinguishing feature. And for 2k, does this result change if you include all cells in clusters 0/2, irrespective of ACTA2 expression? What was the threshold to be deemed positive for ACTA? Is cluster 3 included here?

As described above (reviewer 1; major issue 1), we have revised the figures to examine fibroblast heterogeneity across all datasets (figure 2) and as such this figure is no longer included in its original form. ACTA2 expression in mural cells and fibroblasts is now shown in Figure 1d and ACTA2 expression across the fibroblast subpopulations is now shown in S3a. These figures address the point raised by the reviewer and clarify that ACTA2 expression is detected across both CAFs and mural cells; and that (as stated by the reviewer) examining ACTA2 expression alone is not a sufficiently distinguishing feature for identifying all myofibroblasts/myoCAF in the context of whole tissue analysis, we also explicitly make this point in the results section (lines 155-157).

2j/m – please specify on the legend that this image represents a stroma-rich tumour (if that is the case)

Appropriate additions to the figure legends for all mxIHC panels have now been included in the revised manuscript.

2l – explain in the results or discussion the implications of the non-distinguishing presence of SERPINE1 (normal and tumour expression alike) to its use as a CAF stratifier. Although 2f shows the UMAP and 2l shows the comparison across sites by IHC, is there space to also show its expression on a violin plot?

As described above (reviewer 1; minor issue 3), following additional batch correction processes we have found that the SERPINE1 subpopulation identified in our initial analysis is a transient population and we show that stress response signalling genes are associated with a gene module transiently expressed during myofibroblast transdifferentiation (Figure 4 and associated results section).

Line 226 – to “estimate” functional properties

Line 282 – specify in which samples (normal, tumour, or both)

Line 291-292 – check grammar

These issues have been corrected in the revised manuscript.

Figure 3 – is it possible to predict a “shift” from the telocyte population in health to the expansion of the stress-response population in tumours (are they pre-cursors, for example)?
Check caps in F

We thank the reviewer for this really interesting question. To address this point we have included trajectory inference analysis in our revised manuscript (Figure 4). This analysis demonstrates exactly as the reviewer suggested that the (now termed) adventitial fibroblast population are likely progenitors for fibroblasts undergoing a stress response and furthermore that this is a transient phase in the process of myofibroblast transdifferentiation. Our analysis shows that that all myofibroblast

precursors significantly increase the expression of stress response genes during this process (Figure 4i) but it does also suggest that this is highest and most significantly increased within the adventitial subpopulation (Figure 4i).

S3d/e and S4 – “myoCAF” signature, which is what was used earlier in text (instead of myCAF)

We have corrected inconsistencies in the use of these terms throughout the manuscript.

S3c – interesting lack of functional enrichment in LUSC. In health there seems to be clear functions, in LUAD the two CAF populations have real ECM features, but in LUSC, undistinguished.

We agree with the reviewer that this is a really interesting observation and future studies will look to elucidate the precise mechanisms that underlie these subtype specific differences. To investigate these differences further in our revised manuscript we have included an examination of whether differentially expressed genes could be consistently identified (across patients) between myofibroblast from LUAD and LUSC (Figure S6b). In this analysis we found no significant DEGs, suggesting that the phenotype of the fibroblasts in these tumours is not significantly different. Therefore, we have also used the discussion section to propose that the observed differences between subtypes are likely due to either inherent differences in the tumour phenotypes or because LUSC tumours are near ubiquitously myofibroblast-rich precluding their utility as prognostic biomarkers (page 22 and para 2).

Lines 335-340 – do these cells have an inflammatory signature across sites as was observed in the lung in this study? Because of the strong interest in inflammatory fibroblasts, their ubiquity, and abundance in healthy tissue, and a pretty consistent marker (CD34) is anticipated to be of interest and worth emphasizing.

Based on our trajectory analysis we conclude that the inflammatory fibroblast phenotype is limited to a subset of the adventitial (CD34+) population and so would not propose that this marker could be used in isolation to identify these cells.

Line 343 – check semicolon – emphasize the interpretation here (e.g. indicating something special about cancer)

As suggested, we have added a figure demonstrating the increased probabilities associated with label transfers between myofibroblast across cancer samples compared to IPF (Fig. S5e).

Figure 4i, is there not a control? Seeing the shift in cell counts from healthy to diseased is very informative and notably absent.

Unfortunately, no HNSCC scRNA-seq studies have examined control tissue samples. To address this, we have included MxIHC analysis of these cells across the different tissue types confirming that in all cases the adventitial fibroblasts are increased in control compared to tumour tissues and vice versa for myofibroblasts (Figure 5d-f).

Figure S4b legend, re-specify this is from the PDAC data (assuming it is)

This figure is no longer included in the revised manuscript

Figure 5 - Associated text tells a relevant and interesting narrative but the legend needs improvement (and/or asterisk placement/box plot organisation) to understand the comparisons and where the important enrichments lie (maybe use of letters/symbols is needed?). It is felt that the reader needs to refer back to the text in the main body of the manuscript to appreciate this. B is somehow easier to follow in the absence of text explanation.

As described above, in response to reviewer 1 (major issue 6 and minor issues 15-19), these figures have been significantly reworked in the revised manuscript, which we hope addresses these issues in following the narrative and interpreting the figures.

437 – specify “dampen”?

483 – citation needed for comparison between alveolar and papillary.

These sentences are no longer included in the revised manuscript

Figure 6 – very nice data (could potentially be more space-efficient if needed). Is there a threshold of abundance needed for this prediction (thinking back to the high variability in proportions in Figure 1)?

The reviewer raises an interesting point. It is unclear however whether the threshold referred to in this comment is referring to A) The relative abundance of individual fibroblast subpopulations (as a fraction/percentage of all fibroblasts) within a tumour, or B) the total abundance of fibroblasts.

In reference to A) We have performed the survival analysis using the percentage of all fibroblasts present within a given sample as our metric. This analysis has been performed both using relative abundance of each population as a continuous variable in Cox regression analysis (Figure S6a) showing there is a continuous link between alveolar and myofibroblast abundance and survival rates. We have also applied a threshold to dichotomise samples (as high and low). In our revised manuscript we have provided greater detail on the approach used to define these thresholds shown in Figure 6.

In reference to B) We suspect that the high degree of variability in the absolute abundance identified within the scRNA-seq dataset, particularly with regard to samples having very low numbers of fibroblasts, is the result of technical factors specific to scRNA-seq analysis (likely associated with tissue disaggregation). When analysing whole tissue sections by MxIHC or bulk tissue samples by CIBERSORTx we found fibroblasts were detected as a significantly higher percentage compared to scRNA-seq samples (please see figure below), which is consistent with what has been described previously (PMID: 29988129, 31270426).

583 – or maybe this is due to their relative lack of commitment to a function (as indicated by the spider plots) – this point is linked to the one above on Figures S3c

As described above we have elaborated further on potential explanations for the subtype specific differences observed between LUSC and LUAD.

624 – Consider: can “have” an inflammatory phenotype (rather than develop) Paragraph starting 632 – as mentioned above, please mention the implications of SERPINE1 also being expressed in non-tumour fibroblasts.

These comments are no longer applicable in the revised manuscript.

Reviewer #4, fibroblast characterisation and single cell sequencing
(Remarks to the Author):

Reviewer comments to Hanley et al. Single-cell analysis reveals prognostic fibroblast subpopulations linked to molecular and immunological subtypes of lung cancer.

General comments:

Overall, the study is well designed and data analysis is solid, applying state-of-the-art techniques. The authors use single-cell transcriptomic data for the deconvolution of clinical bulk RNA-seq datasets from lung cancer patients, and thereby provide important information about the cellular composition of the patient samples. However, considering the depth of single-cell data, it feels as if the analysis, as presented in the manuscript remains rather generalized. The authors use plenty of publically available databases/-sets which at least for this reviewer were difficult to follow and understand the rationale behind which dataset(s) were chosen for the respective analysis. While the results are presented coherently in each individual figure, it is less clear how the wealth of results are connected to each other.

We would like to thank the reviewer for their kind comments on the application of state-of-the-art techniques in our manuscript. In the revised version of our manuscript we hope to have put our results in a more coherent structure with increased validation of each result presented to provide a clearer overview of how these results connect to each other and enhance our understanding of fibroblast heterogeneity in NSCLC.

Specific		comments
Regarding	Figure	1-2:
1. It is unclear why the authors choose to do the fibroblast subtype analysis on their own scRNA-seq dataset, and not directly using the integrated dataset.		

As described above (reviewer 1; major point 1), we agree with the reviewer that the original structure of our manuscript was not as clear as it could be and have now amended this to perform all analysis of fibroblast heterogeneity on the integrated dataset (revised Figure 2).

2. Further, regarding the usage of SERPINE1 as a marker for a specific CAF subset the data are somewhat contradictory between the antibody staining and the scRNA-seq results. The authors propose SERPINE1 as a marker for their C2 subset mainly derived from LUSC samples (Fig 2a-b), however in their tissue analysis the SERPINE1+ population is found to almost equal amounts in the both LUAD and LUSC (Fig 2k-l). One reason for this can be, that cells in C2 are derived from almost exclusively one sample (Fig S2a), which decreases the validity of C2 cells as a representative subtype. Additionally, both markers, POSTN and SERPINE1 (protein: PAI-1), that are used to determine cell subset tissue distribution are secreted proteins, and thus less appropriate to identify the producer cell. The authors should consider to use marker proteins that are intracellularly or membrane bound. Alternatively, multiplexed in situ hybridization (e.g. RNAscope) could be employed (as validation).

We agree with the reviewer that the marker selection for our original MxIHC was suboptimal and in the revised manuscript we have defined an improved MxIHC panel ensuring that the markers used are consistent between RNA-seq and protein detection and that they have previously been shown

to be detected intracellularly. This approach is described in the revised results (page 13 and para 1; Figure 3).

3. In Fig S2f, the authors should explain how can it be that the population of SMA+SERPINE1+ cells is found in regions where no SMA+ cells are highlighted (density plot, compare first image and last image in second row)?

These heatmap plot showed the area where density is highest, so the “negative” regions represent relatively lower density rather than absence of cells completely. We appreciate the reviewers point on how this may not be the clearest method for displaying these data and have replaced the equivalent figures with down sampled point pattern plots in the revised figures (e.g. Figure 3c and Figure 7a).

4. The seventh LUSC sample contributes only marginally to the tumor specific clusters, but instead to a comparable amount as the normal tissue samples to cluster C1 and C4. Should this sample still be considered as tumor sample?

All tumour samples were confirmed as such by a pathologist during sample collection. We believe this to be more definitive for determining the “type” of sample rather than the fibroblast phenotypes isolated from it.

Regarding Figure 3:

5. As mentioned above, it is surprising that the authors do not use the fibroblast subpopulation result from their meta-analysis for marker gene selection and subsequent tissue analysis. Further, it is unfortunate that the authors keep their single cell analysis restricted to the four major subpopulation clustering (alveolar, telocytes, myofibs and stress). For example, all four subpopulations contain cells collected from normal as well as tumor samples (Fig. 3c) and the intra-cluster differential expression between normal fibroblasts and tumor-derived fibroblast would be relevant analysis. In relation to these data, how do the authors reason about the very low amount of alveolar FB from LUSC samples, what is the underlying cause?

We thank the reviewer for raising this interesting point about how the data could be used to yield a deeper understanding of fibroblast phenotypes in these tissues. In our revised draft, as suggested, we have used the integrated dataset for fibroblast marker identification (Table S2) and defining the MxIHC panel (Figure 3a). We have also performed intra-cluster analyses between tumour and control tissue (Figure 2h-j, S2g-h and Table S5) and between NSCLC subtypes (Figure S6b). Furthermore, we have now incorporated trajectory inference analysis (Figure 4) to further examine more subtle changes in gene expression, in addition to those associated with fibroblast subpopulations. We have also used our MxIHC data to examine possible reasons for why the alveolar fibroblasts are infrequently found in LUSC samples, showing that this is at least partially due to field effects extending beyond the tumour border (Figure S3f-h).

6. The authors should perform additional analysis, such as pseudo-time that have the potential reveal the relationships of the defined subpopulations, intermediate cell stages as well as relevant differential gene expression pattern along the trajectory.

We have included this analysis in the revised Figure 4.

7. Further analysis beyond Gene Ontology categories or matrix gene expression levels within the major subpopulations should be considered by the authors to increase the impact of the manuscript. It becomes more and more appreciate that fibroblasts and CAF exhibit a high level of inter- as well as intra organ heterogeneity, and thus the somewhat general analysis considering only the major subpopulations misses out on potentially important and relevant details. For reference on fibroblast heterogeneity see: Tallquist M, doi.org/10.1146/annurev-physiol-021119-034527, LeBleu & Neilson, FASEB J DOI: 10.1096/fj.201903188R, and Sahei et al. Nature Reviews <https://doi.org/10.1038/s41568-019-0238-1>.

This heterogeneity has been more comprehensively assessed by the analyses described in response to the previous two comments. Furthermore, we have also compared the phenotypes that we describe in lung tissue to other fibroblast phenotypes using gene set variation analysis (Figure S2f) and performed further cross-tissue analysis than originally presented (Figure 5).

8. It is unclear what purpose the comparison to the myCAF and iCAF signatures fulfills. Further, all genes included in myCAF and iCAF signature should be disclosed for proper interpretation of the analysis presented in Fig S3d-e.

As described for the point above the analysis of these signatures was performed to provide context to the phenotypes described by other studies. We have now expanded this analysis to include multiple signatures (Figure S2f) and these are detailed in Table S4.

Regarding Figure 4:

9. The authors employ machine learning algorithm to predict the four identified major fibroblast subpopulations in other datasets, which is an interesting aspect and could possibly developed into an applicable tool for scRNA-seq analysis. However, to test the functionality or their prediction module, the authors use only three different datasets (one from lung and two from pancreatic or head and neck tumors), and only analyze the preselected fibroblast populations of the test datasets. Thereby, important controls to interpret the results are not provided. It would be important to see the predictive score for unrelated cell types, such as epithelial cells or endothelial cells. Further, closer related connective tissue cell types, such as mural cells (pericytes, smooth muscle cells) or chondrocytes should be analyzed as control groups to validate the algorithm. Generally, the data presented in Figure 4 and S4 does somehow lack a proper connection to the other results and the authors could consider to omit the data in this manuscript.

As reviewers 2&3 described this section as a valuable addition to the manuscript, we have chosen to keep it in the revised version. However, we have now performed this analysis more thoroughly expanding to colorectal cancer and supplementing the original scRNA-seq analysis with mxIHC. We agree with the reviewer that designing a machine learning or AI approach to distinguish between all cell types within any tissue sample would have useful applications, but this is beyond the scope of this study. In the approach that we have implemented the machine learning model is trained on pre-selected fibroblasts, and this pre-selection is therefore a key pre-requisite for its implementation.

10. The statement that telocytes (CD34+ fibroblasts) and myofibroblasts are conserved across

tissues is likely an overstatement, especially considering the limited amounts of tissues tested and the lack of proper controls to validate the efficacy of the prediction algorithm.

As described for the point above this analysis has now been expanded to additional tissue types confirming the existence of these two populations in multiple tissues and confirming that the phenotype is largely conserved. In order not to overstate these results we have now toned down the language used to describe these conclusions to include a caveat that they are conserved across the tissues analysed.

11. In relation, Fig 4e depicts predicted myofibroblasts in many clouds (clusters) of the PDAC UMAP landscape, which can be a sign of an imprecise categorizations, since the cells distributed to the different clouds in the UMAP landscape should exhibit specific molecular fingerprints. The visualization of the clustering result from the different datasets (IPF, PDAC, HNSCC), overlaid with the prediction would be informative as to what extent the prediction recapitulates the dataset-specific cluster determination.

We appreciate the reviewer's point and agree that this is a relevant consideration. We have now added a UMAP plot with associated tissue type information overlaid to represent dataset specific traits (Figure 5a). We propose this as an alternative to the suggestion of showing a clustering solution because there are multiple approaches and parameters that can influence the result of cell clustering and therefore, we felt that the tissue type would be more definitive for assessing the different classifier results.

12. Additionally, it would be helpful for the interpretation to be able to see the genes that are used by the algorithm for the subtype prediction.

The genes used are those shared between the 1422 features used for fibroblast integration and those genes detected in each query dataset (i.e. some genes may be omitted if not detected in a particular query dataset). The full list of 1422 genes used for dataset integration and cross tissue analysis are now provided in Supplementary Data 10.

Related to Figure 5:

13. The authors use scRNA-seq data for deconvolution of bulk RNA-seq patient data from the TCGA, using the online tool CIBERSORTx. This is an interesting analysis with substantial high impact. However, it is surprising that the authors choose to use two datasets from other publications to generate the signature matrix which serves as the bases for the deconvolution, instead of their (whole) meta-analysis dataset; please explain. For proper interpretation of the deconvolution, it is necessary that the authors reveal the genes that are included in the signature matrix (Fig S5a).

The rationale for dataset choice when building the signature matrix was to only use scRNA-seq data generated using the Smart-Seq2 platform. This is to avoid the technical variation generated between scRNA-seq and bulk transcriptome data when droplet-based technologies are used for scRNA-seq. The original CIBERSORTx study (PMID: 31061481) demonstrate this issue and showed that SMART-Seq2 data had less technical variation compared to the droplet-based technologies. Furthermore, in our simulation experiments to optimise the deconvolution algorithm we found that using only SMART-Seq2 data significantly improved overall accuracy compared to using all data from droplet and plate based scRNA-seq approaches.

14. After successful decoding of fibroblast subtype abundance in clinical tumor samples, the authors correlate the prevalence of each fibroblast subpopulations with a plethora of clinical parameters for both LUAD and LUSC samples. Unfortunately, it is difficult to comprehend the message from the results presented in Fig 5, other than that normal fibroblast populations (alveolar & telocyte) have a high abundance in control samples, and CAF subpopulations (myofibroblasts & stress) have a high abundance in tumor samples. It is unclear what relevance the statistical significance calculated in Fig 5a-b has, since (almost) all of the data groups are “significant”, please explain and revise. Please also indicate the number of samples (n) that are included in each block and used to calculate statistics.

We thank the reviewer for pointing out this difficulty in interpreting the data presented in this figure. As described above, in response to reviewer 1 (major issue 6 and minor issue 15), we have now amended this figure to make the statistical comparisons clearer and ensure that they are performed considering comparisons across and within relevant subtypes separately (now shown in Figure 3 and Figure 7).

15. Is this seemingly low difference of the two CAF subpopulations due to a too general classification (see comments above)?

In our revised manuscript, as described above in relation to the comment from reviewer 1 (additional issue 3), we have applied a more robust approach to removing batch effects between samples and as a result of this alteration in data processing we have now identified myofibroblasts as the principal “CAF” population as opposed to the two CAF subpopulations identified in our original manuscript.

16. Additionally, an analysis focusing on the difference of CAF subpopulations between the two different tumor types (LUAD & LUSC) has the potential to unveil important transcriptomic features within CAFs and raise the impact of the manuscript (see also comments below).

We have now included a differential expression analysis conducted between myofibroblasts isolated from LUAD and LUSC (Figure S6b). this analysis showed that we could not identify any differentially expressed genes between these subtypes. This is also concordant with the inter-cancer type analysis, which showed that there is high concordance in the myofibroblast phenotype between all cancer types that we have analysed. Therefore, we now suggest that the differences observed between LUAD and LUSC in terms of survival correlations are potentially due to inherent differences in the tumour cells from these subtypes, which may affect their ability to respond to myofibroblast’s tumour-promoting properties (discussion section, page 22 para 2).

17. Similar to the Gene Ontology analysis, the network analysis presented in Fig 5c-d has a generalized character and the interpretation of the results is difficult. For example, what does the correlation of LUSC myofibroblasts with the “intermediate filaments and keratin” module mean, are the myofibroblasts suggested to express genes of the module, or does this allow conclusions about the overall tissue composition?

This point was similarly raised by reviewer 1 (major issue 6) and therefore we have removed these network analyses results from our revised manuscript and focused on more clinically relevant sample traits.

18. However, the analysis does reveal a difference in the “cell cycle” correlation between myofibroblasts from LUAD and LUSC, respectively. The authors do not follow up on this, but could probably investigate this in more detail.

As described above we no longer include this network analysis. However, we have now clearly demonstrated a link between myofibroblasts and poorly differentiated tumours (Figure 7a-d), which represent increased numbers of proliferating tumour cells, therefore confirming a similar relationship between these cells and cell cycle changes originally inferred from transcriptomic analysis. We also agree with the reviewer that this relationship does warrant further investigation and will be the focus of future functional studies.

Regarding Figure 6:

The clinical relevance lies outside of this reviewers’ competence. Thus, the comments should be regarded as from a non-expert on the clinical aspect.

19. The authors apply their fibroblast abundance scores to correlate with patient outcomes (overall survival). This analysis illustrates the power of scRNA-seq data and contains highly relevant and impactful results. It is interesting that only ratios between myofibroblasts and alveolar fibroblasts exhibit a prognostic relevance for LUAD patients. Obvious questions are, why the prognostic capacity is limited to LUAD and not present in LUSC as well, and why the stress-fibroblasts do not exhibit any prognostic capability. Can the authors comment, as well as investigate possible factors underlying these effects of LUAD myofibroblasts on overall survival?

As described above, our analysis has demonstrated that these differences are unlikely to be due to phenotypic differences within the fibroblast subpopulations across NSCLC subtypes. Instead, we have two hypotheses for why differences are observed in survival correlations between the two NSCLC subtypes (described in the discussion section page 22 para 2).

- 1) *LUSC tumours are near ubiquitously myofibroblast-rich and it is possible that the lack of heterogeneity observed in the stromal phenotype associated with these tumours limits their efficacy in stratifying patient survival rates.*
- 2) *Variation in the phenotype of LUSC malignant epithelial cells may render these cells less capable of responding to tumour-promoting cues from myofibroblasts, thereby reducing the impact of stromal cells on disease progression.*

20. The authors should control that these effects are independent of the overall tumor burden/stage (e.g. high burden = high number of myofibroblasts, low burden = high number of alveolar fibroblasts), or other parameters (multivariate analysis).

We have performed a multivariate cox regression analysis to demonstrate that both the myofibroblasts and alveolar fibroblast correlations with overall survival rates are independent of disease stage and patient age (Figure 6i-j and S6c-d)

21. Further, since the authors have the single-cell transcriptional data available, an analysis of the LUAD myofibroblast population on the background of correlation to patient survival could reveal important gene expression patterns and would increase the impact of the manuscript. Would a

more in-depth analysis (see comments above) of the myofibroblasts (and other) subpopulation lead to more defined prognostic model?

As described above, we examined phenotypic differences between myofibroblasts isolated from control and tumour samples and found tumour-specific upregulation of interstitial collagens (Figure 2h) as well as other specific genes (Figure S2g-h and table S5). Suggesting that this may be a key phenotypic change associated with their role in tumour progression. However, these differences were not substantial enough to enable the CIBERSORTx analysis to accurately differentiate between these further sub-classifications precluding an analysis of whether this could improve the prognostic model.

22. Since the stratification of patients into prognostic groups is suggested by the authors, from a clinically perspective, it is important to understand at what time the prognostic correlation can be performed, could the authors comment on this. Finally, what additional prognostic analysis could be introduced in accordance to the results from the myofibroblast-correlation?

We show that fibroblast subpopulation abundance has prognostic value in samples analysed at surgical resection, which is the typical source of the material provided for the bulk transcriptomic datasets analysed. To determine the prognostic capacity of these biomarkers in eg. biopsy tissue is beyond this is beyond the scope of this study.

Minor comments to the Figures:

23. labeling in Fig S2c should be changed to (C2)

24. labeling in Fig 3f is cut, please correct

25. in Fig 6e, f, g, add legends for bar plot segments

26. in Fig S6, why different order of spider plots between a & b?

27. generally, the authors could give more information in the Figure legends, especially in the legends of the supplementary figures, since they are not similarly limited in word counts

These minor comments have been addressed in the revised manuscript.

REVIEWER COMMENTS

Reviewer #1 (Remarks to the Author):

The extensive re-analysis and revision is appreciated, but there remains outstanding issue that center around the use of single marker (CD34) for adventitial fibroblasts. In pathology, CD34 is well known as a marker of endothelial cells, and this is supported by the staining shown in Figures 3b/Supplemental DATA 1. Also, in Supplemental DATA3, panel e, the structure that show CD34 staining appears to be vascular endothelium. Higher magnification of Fig.3b (middle panel) focusing on the CD34 stained areas would be helpful. Nevertheless, a second marker for adventitial fibroblasts (and also for alveolar fibroblasts) is essential. Perhaps AKAP12 or APOD might serve the role of second marker for interstitial fibroblasts if it is not expressed in the alveolar and myofibroblasts. The need for second marker is also suggested with the use of POSTN and ACTA2 as markers of myofibroblasts. Supplementary Figure S3, panel b appears to show that there is heterogeneity of staining for these 2 markers for myofibroblasts.

The second issue is the conclusion that there is a difference in the abundance of adventitial and alveolar fibroblasts in LUAD compared to LUSC, and the abundance of these 2 fibroblast types is more heterogeneous in LUAD than LUSC. Figure 3c shows that alveolar and/or adventitial fibroblasts are found mainly at the edges or peripheral areas of the tumor, and these are areas in NSCLC tumors, especially LUAD where non-neoplastic alveolar tissue can be entrapped. Therefore, the differences in abundance of various fibroblast populations observed in Fig.3e could be due to the inclusion of these non-neoplastic lung tissue in the tumors measured. For MxIHC, it could potentially be corrected if the analysis is strictly restricted to tumor areas that do not have entrapped non-neoplastic lung alveolar tissue. What has been done to alleviate this concern?

The finding that adventitial and alveolar fibroblasts are less abundant in grade 3 LUAD is interesting. However, as mentioned, G3 includes both solid and micropapillary predominant LUADs. While these 2 subtypes are associated with poor prognosis, they are morphologically very different, with micropapillary types retaining some alveolar structures. Therefore, the correlation for micropapillary and solid types should be separately analyzed.

As the pathologist coauthor may be aware of, most LUADs are “mixed type” with multiple patterns, and the classification (and grading) is based on the predominant (most abundant) pattern observed. Therefore, a plot of fibroblast subtype percentages according to the percent of various LUAD patterns (lepidic, acinar/papillary, micropapillary and solid) can be highly informative.

In term of prognostic association of different fibroblast types, the differences between myofibroblasts vs. alveolar/adventitial fibroblasts could be related to the LUAD subtypes, as grade is also correlated

with both poor prognosis and fibroblast types. Therefore, the multivariate analysis should include adjusting for the tumor grade and histology subtype.

While pseudotime and trajectory inference analyses may suggest a “transdifferentiation” event, but this in silico analysis result is purely for hypothesis generating, and biological validation (e.g. using cultured fibroblasts) is essential to prove the robustness of the methodology.

In Figure 5d, the histology images of the tumors are so small that it is not possible to assess them properly, thus not possible to assess the spatial distribution of fibroblast subpopulations. Frankly this data is not very relevant to the main topic of the paper, as stated in the manuscript title, thus does not add value to the paper.

Minor suggestion:

When mural cells is first mentioned in the introduction (line 116), perhaps adding (vascular smooth muscle cells and pericytes) may clarify what it means.

Reviewer #2 (Remarks to the Author):

The revised manuscript is much approved. The new manuscript does a much better job of integrating findings from the scRNA-seq data in the early parts of the manuscript with the broader analyses using multiple additional public datasets. Addressing batch effects in the scRNA-seq data seems to have improved the inconsistencies in the original presentation and the approach for discriminating fibroblasts and mural cells is quite useful. The high degree of fibroblast heterogeneity is better characterized, the enrichment of myofibroblasts in tumor is better supported, and the multiplexed imaging provides additional useful supporting data. The mxIHC data was applied to samples outside of the scRNA-seq cohort, the markers were well selected, and the interpretation appropriate, leveraging the ciphersort analysis and the spatial information to investigate fibroblast population distributions in LUSC and LUAD. The new model of fibroblast activation using trajectory inference analysis is an interesting finding, as are the associations with myofibroblasts and cell stress. The links to heat shock stress response is interesting and references have been adjusted to cite prior work in that area.

Reviewer #3 (Remarks to the Author):

As mentioned in the first review, this study represents an important addition to the growing field about fibroblast heterogeneity. Many suggestions and concerns were raised by the four reviewers following the initial submission, however, it is felt that these have been satisfactorily addressed, both through explanations in the rebuttal letter as well as in the revisions to the manuscript itself.

Reviewer #4 (Remarks to the Author):

The authors have ambitiously revised their manuscript in response to the suggestions made by me and the other reviewers. I have nothing more to add and think that the paper can now be accepted.

Point-by-point response to the reviewers' comments:

We would like to thank all the reviewers for their expert insight and constructive feedback on our manuscript.

Reviewer #1 (Remarks to the Author):

The extensive re-analysis and revision is appreciated, but there remains outstanding issue that center around the use of single marker (CD34) for adventitial fibroblasts. In pathology, CD34 is well known as a marker of endothelial cells, and this is supported by the staining shown in Figures 3b/Supplemental DATA 1. Also, in Supplemental DATA3, panel e, the structure that show CD34 staining appears to be vascular endothelium. Higher magnification of Fig.3b (middle panel) focusing on the CD34 stained areas would be helpful. Nevertheless, a second marker for adventitial fibroblasts (and also for alveolar fibroblasts) is essential. Perhaps AKAP12 or APOD might serve the role of second marker for interstitial fibroblasts if it is not expressed in the alveolar and myofibroblasts. The need for second marker is also suggested with the use of POSTN and ACTA2 as markers of myofibroblasts. Supplementary Figure S3, panel b appears to show that there is heterogeneity of staining for these 2 markers for myofibroblasts.

We thank the reviewer for acknowledging the extensive re-analysis we have performed in response to the original round of peer review. In response to the request for using additional markers to identify the fibroblast subpopulations described in our manuscript we direct the reviewer to page 13 paragraph 1 and page 27 "Stromal cell identification and Histo-Cytometry", where we describe the design and implementation of our MxIHC panel. This details the 7 markers that were used in combination to identify each cell population. This panel included Pan-cytokeratin, CD31, MCAM, AOC3, CD34, POSTN and ACTA2. As the reviewer rightly points out, the use of CD34 alone would not be sufficient to identify adventitial fibroblasts as this marker is known to be expressed by endothelial cells. In our manuscript we identify adventitial fibroblasts as PanCK- CD31- MCAM- CD34>AOC3/POSTN/ACTA2. This precludes the mis-identification of CD31+ endothelial cells as adventitial fibroblasts. To demonstrate this point direct the reviewer to re-examine the panel they mention (Supplemental DATA 3e) and note that whilst they are correct to point out that CD34 staining is observed in vascular endothelium; there are no adventitial fibroblasts detected in that region of interest (Figure 3d, because these cells are also positive for CD31 (CD31+CD34+ [endothelial] cells). The heterogeneity observed between POSTN and ACTA2 as myofibroblast markers is also to be expected based on our scRNA-seq data. *POSTN* was identified as the optimal IHC marker for myofibroblasts (Figure 3a), however was also found to be variably expressed between tumour-associated and control tissue myofibroblasts ($p=9.41e7$, $adj.p=0.07$; Table S5). This motivated us to also use the well described myofibroblast marker ACTA2 to ensure all these cells would be detected by our mxIHC panel.

The second issue is the conclusion that there is a difference in the abundance of adventitial and alveolar fibroblasts in LUAD compared to LUSC, and the abundance of these 2 fibroblast types is more heterogeneous in LUAD than LUSC. Figure 3c shows that alveolar and/or adventitial fibroblasts are found mainly at the edges or peripheral areas of the tumor, and these are areas in NSCLC tumors, especially LUAD where non-neoplastic alveolar tissue can be entrapped. Therefore, the differences in abundance of various fibroblast populations observed in Fig.3e could be due to the inclusion of these non-neoplastic lung tissue in the tumors measured. For MxIHC, it could potentially be corrected if the

analysis is strictly restricted to tumor areas that do not have entrapped non-neoplastic lung alveolar tissue. What has been done to alleviate this concern?

The mxIHC analysis presented in our revised manuscript was performed precisely as the reviewer described, excluding entrapped non-neoplastic tissue additionally found within the tissue section stained (as illustrated in Figure 3c). This showed that myofibroblasts represented a significantly higher proportion (and therefore alveolar/adventitial fibroblasts a lower proportion) of the fibroblasts found in LUSC than in LUAD, when specifically analysing tumour tissue and ensuring non-neoplastic tissue was discounted. To clarify this we have amended paragraph 1 on page 14 and the legend for figure 3.

The finding that adventitial and alveolar fibroblasts are less abundant in grade 3 LUAD is interesting. However, as mentioned, G3 includes both solid and micropapillary predominant LUADs. While these 2 subtypes are associated with poor prognosis, they are morphologically very different, with micropapillary types retaining some alveolar structures. Therefore, the correlation for micropapillary and solid types should be separately analyzed. As the pathologist coauthor may be aware of, most LUADs are “mixed type” with multiple patterns, and the classification (and grading) is based on the predominant (most abundant) pattern observed. Therefore, a plot of fibroblast subtype percentages according to the percent of various LUAD patterns (lepidic, acinar/papillary, micropapillary and solid) can be highly informative.

We have performed this more detailed analysis which has highlighted that increased myofibroblast abundance in G3 tumours correlates with solid morphologies (but not micropapillary). These data are presented in the revised Figure S7b and c and described in page 19 paragraph 2.

In term of prognostic association of different fibroblast types, the differences between myofibroblasts vs. alveolar/adventitial fibroblasts could be related to the LUAD subtypes, as grade is also correlated with both poor prognosis and fibroblast types. Therefore, the multivariate analysis should include adjusting for the tumor grade and histology subtype.

We have performed the multivariate analysis adjusting for grade and this shows that both myofibroblasts and alveolar fibroblasts remain independently prognostic across the entire cohort (please see figures below). However, because information pertaining to grade is only available from two of the four cohorts analysed this markedly reduces the sample size available for survival analysis. Therefore, in the revised manuscript we have now made reference to the results presented below (page 19 paragraph 1) but opted to keep the full 4 cohort survival analysis and statistics presented in figure 6 i/j as per the revised manuscript. However, we can include this panel in Figure S6 if required.

While pseudotime and trajectory inference analyses may suggest a “transdifferentiation” event, but this *in silico* analysis result is purely for hypothesis generating, and biological validation (e.g. using cultured fibroblasts) is essential to prove the robustness of the methodology.

We agree with the reviewer that further biological validation of our trajectory analysis results would be highly informative. However, this has many potentially confounding issues associated with artificial phenotypic skewing generated by *in vitro* culture methods, which makes this degree of validation beyond the scope of our current manuscript. This validation and line of enquiry will be the focus of future research studies and we hope that our current manuscript will enable multiple research groups to work together or in parallel to tackle this challenging problem.

In Figure 5d, the histology images of the tumors are so small that it is not possible to assess them properly, thus not possible to assess the spatial distribution of fibroblast subpopulations. Frankly this data is not very relevant to the main topic of the paper, as stated in the manuscript title, thus does not add value to the paper.

We respectfully disagree with this comment. The images provided are of sufficient resolution to examine the TMA cores presented if the reader ‘zooms in’ and we believe that the results presented in this figure provide important context to previous studies investigating fibroblast heterogeneity in other cancer and tissue types.

Minor suggestion:

When mural cells is first mentioned in the introduction (line 116), perhaps adding (vascular smooth muscle cells and pericytes) may clarify what it means.

We have amended this accordingly.

Reviewer #2 (Remarks to the Author):

The revised manuscript is much approved. The new manuscript does a much better job of integrating findings from the scRNA-seq data in the early parts of the manuscript with the broader analyses using multiple additional public datasets. Addressing batch effects in the scRNA-seq data seems to have improved the inconsistencies in the original presentation and the approach for discriminating fibroblasts and mural cells is quite useful. The high degree of fibroblast heterogeneity is better characterized, the enrichment of myofibroblasts in tumor is better supported, and the multiplexed

imaging provides additional useful supporting data. The mxIHC data was applied to samples outside of the scRNA-seq cohort, the markers were well selected, and the interpretation appropriate, leveraging the ciphersort analysis and the spatial information to investigate fibroblast population distributions in LUSC and LUAD. The new model of fibroblast activation using trajectory inference analysis is an interesting finding, as are the associations with myofibroblasts and cell stress. The links to heat shock stress response is interesting and references have been adjusted to cite prior work in that area.

Reviewer #3 (Remarks to the Author):

As mentioned in the first review, this study represents an important addition to the growing field about fibroblast heterogeneity. Many suggestions and concerns were raised by the four reviewers following the initial submission, however, it is felt that these have been satisfactorily addressed, both through explanations in the rebuttal letter as well as in the revisions to the manuscript itself.

Reviewer #4 (Remarks to the Author):

The authors have ambitiously revised their manuscript in response to the suggestions made by me and the other reviewers. I have nothing more to add and think that the paper can now be accepted.